# YOU ARE ALLSET: A MULTISET LEARNING FRAMEWORK FOR HYPERGRAPH NEURAL NETWORKS

**Eli Chien**[*]    **Chao Pan**[*]    **Jianhao Peng**[*]    **Olgica Milenkovic**
Department of Electrical and Computer Engineering
University of Illinois, Urbana-Champaign
`{ichien3,chaopan2,jianhao2,milenkov}@illinois.edu`

## ABSTRACT

Hypergraphs are used to model higher-order interactions amongst agents and there exist many practically relevant instances of hypergraph datasets. To enable the efficient processing of hypergraph data, several hypergraph neural network platforms have been proposed for learning hypergraph properties and structure, with a special focus on node classification tasks. However, almost all existing methods use heuristic propagation rules and offer suboptimal performance on benchmarking datasets. We propose AllSet, a new hypergraph neural network paradigm that represents a highly general framework for (hyper)graph neural networks and for the first time implements hypergraph neural network layers as compositions of two multiset functions that can be efficiently learned for each task and each dataset. The proposed AllSet framework also for the first time integrates Deep Sets and Set Transformers with hypergraph neural networks for the purpose of learning multiset functions and therefore allows for significant modeling flexibility and high expressive power. To evaluate the performance of AllSet, we conduct the most extensive experiments to date involving ten known benchmarking datasets and three newly curated datasets that represent significant challenges for hypergraph node classification. The results demonstrate that our method has the unique ability to either match or outperform all other hypergraph neural networks across the tested datasets: As an example, the performance improvements over existing methods and a new method based on heterogeneous graph neural networks are close to $4\%$ on the Yelp and Zoo datasets, and $3\%$ on the Walmart dataset. Our AllSet network implementation is available online[1].

## 1 INTRODUCTION

Graph-centered machine learning, and especially graph neural networks (GNNs), have attracted great interest in the machine learning community due to the ubiquity of graph-structured data and the importance of solving numerous real-world problems such as semi-supervised node classification and graph classification (Zhu, 2005; Shervashidze et al., 2011; Lü & Zhou, 2011). Graphs model *pairwise* interactions between entities, but fail to capture more complex relationships. Hypergraphs, on the other hand, involve hyperedges that can connect more than two nodes, and are therefore capable of representing *higher-order* structures in datasets. There exist many machine learning and data mining applications for which modeling high-order relations via hypergraphs leads to better learning performance when compared to graph-based models (Benson et al., 2016). For example, in subspace clustering, in order to fit a $d$-dimensional subspace, we need at least $d+1$ data points (Agarwal et al., 2005); in hierarchical species classification of a FoodWeb, a carbon-flow unit based on four species is significantly more predictive than that involving two or three entities (Li & Milenkovic, 2017). Hence, it is desirable to generalize GNN concepts to hypergraphs.

One straightforward way to generalize graph algorithms for hypergraphs is to convert hypergraphs to graphs via clique-expansion (CE) (Agarwal et al., 2005; Zhou et al., 2006). CE replaces hyperedges by (possibly weighted) cliques. Many recent attempts to generalize GNNs to hypergraphs can be viewed as redefining hypergraph propagation schemes based on CE or its variants (Yadati et al.,

---

[*]Equal contribution.    [1] https://github.com/jianhao2016/AllSet

2019; Feng et al., 2019; Bai et al., 2021), which was also originally pointed out in (Dong et al., 2020). Despite the simplicity of CE, it is well-known that CE causes distortion and leads to undesired losses in learning performance (Hein et al., 2013; Li & Milenkovic, 2018; Chien et al., 2019b).

In parallel, more sophisticated propagation rules directly applicable on hypergraphs, and related to tensor eigenproblems, have been studied as well. One such example, termed Multilinear PageRank (Gleich et al., 2015), generalizes PageRank techniques (Page et al., 1999; Jeh & Widom, 2003) directly to hypergraphs without resorting to the use of CE. Its propagation scheme is closely related to the Z eigenproblem which has been extensively investigated in tensor analysis and spectral hypergraph theory (Li et al., 2013; He & Huang, 2014; Qi & Luo, 2017; Pearson & Zhang, 2014; Gautier et al., 2019). An important result of Benson et al. (2017) shows that tensor-based propagation outperforms a CE-based scheme on several tasks. The pros and cons of these two types of propagation rule in statistical learning frameworks were examined in Chien et al. (2021a). More recently, it was shown in Tudisco et al. (2020) that label propagation based on CE of hypergraphs does not always lead to acceptable performance. Similarly to Chien et al. (2021a), Benson (2019) identified positive traits of CE eigenvectors but argued in favor of using Z eigenvectors due to their more versatile nonlinear formulation compared to that of the eigenvectors of CE graphs.

We address two natural questions pertaining to learning on hypergraphs: "Is there a general framework that includes CE-based, Z-based and other propagations on hypergraphs?" and, "Can we learn propagation schemes for hypergraph neural networks suitable for different datasets and different learning tasks?" We give affirmative answers to both questions. We propose a general framework, AllSet, which includes both CE-based and tensor-based propagation rules as special cases. We also propose two powerful hypergraph neural network layer architectures that can learn adequate propagation rules for hypergraphs using multiset functions. Our specific contributions are as follows.

**1.** We show that using AllSet, one can not only model CE-based and tensor-based propagation rules, but also cover propagation methods of most existing hypergraph neural networks, including HyperGCN (Yadati et al., 2019), HGNN (Feng et al., 2019), HCHA (Bai et al., 2021), HNHN (Dong et al., 2020) and HyperSAGE (Arya et al., 2020). Most importantly, we show that all these propagation rules can be described as a composition of two multiset functions (leading to the proposed method name AllSet). Furthermore, we also show that AllSet is a hypergraph generalization of Message Passing Neural Networks (MPNN) (Gilmer et al., 2017), a powerful graph learning framework encompassing many GNNs such as GCN (Kipf & Welling, 2017) and GAT (Veličković et al., 2018).

**2.** Inspired by Deep Sets (Zaheer et al., 2017) and Set Transformer (Lee et al., 2019), we propose AllDeepSets and AllSetTransformer layers which are end-to-end trainable. They can be plugged into most types of graph neural networks to enable effortless generalizations to hypergraphs. Notably, our work represents the first attempt to connect the problem of learning multiset function with hypergraph neural networks, and to leverage the powerful Set Transformer model in the design of these specialized networks.

**3.** We report, to the best of our knowledge, the most extensive experiments in the hypergraph neural networks literature pertaining to semi-supervised node classification. Experimental results against ten baseline methods on ten benchmark datasets and three newly curated and challenging datasets demonstrate the superiority and consistency of our AllSet approach. As an example, AllSetTransformer outperforms the best baseline method by close to $4\%$ in accuracy on Yelp and Zoo datasets and $3\%$ on the Walmart dataset; furthermore, AllSetTransformer matches or outperforms the best baseline models on nine out of ten datasets. Such improvements are not possible with modifications of HAN (Wang et al., 2019b), a heterogeneous GNN, adapted to hypergraphs or other specialized approaches that do not use Set Transformers.

**4.** As another practical contribution, we also provide a succinct pipeline for standardization of the hypergraph neural networks evaluation process based on Pytorch Geometric (Fey & Lenssen, 2019). The pipeline is built in a fashion similar to that proposed in recent benchmarking GNNs papers (Hu et al., 2020a; Lim et al., 2021). The newly introduced datasets, along with our reported testbed, may be viewed as an initial step toward benchmarking hypergraph neural networks.

All proofs and concluding remarks are relegated to the Appendix.

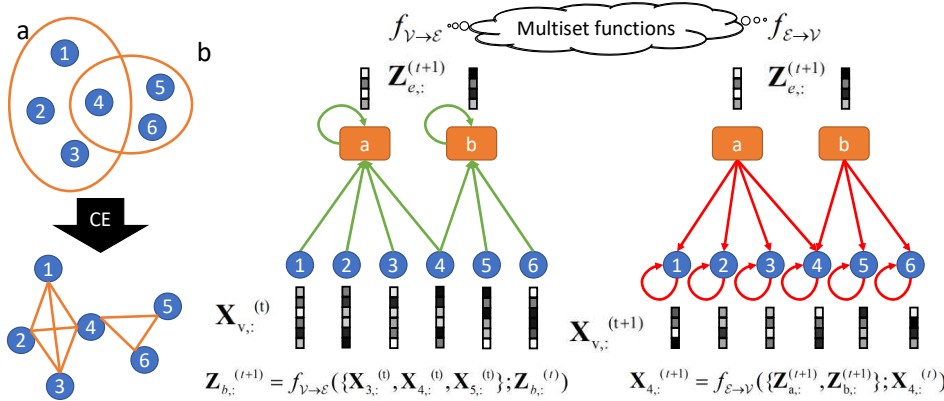

Figure 1: Left: The difference between a hypergraph and clique-expanded graph. Right: Illustration of our AllSet framework for the hypergraph depicted on the left. Included is an example on the aggregation rule for hyperedge $b$ and node 4. The key idea is that $f_{\mathcal{V} \to \mathcal{E}}$ and $f_{\mathcal{E} \to \mathcal{V}}$ are two multiset functions, which by definition are permutation invariant with respect to their input multisets.

## 2 BACKGROUND

**Notation.** A hypergraph is an ordered pair of sets $\mathcal{G}(\mathcal{V}, \mathcal{E})$, where $\mathcal{V} = \{1, 2, \ldots, n\}$ is the set of nodes while $\mathcal{E}$ is the set of hyperedges. Each hyperedge $e \in \mathcal{E}$ is a subset of $\mathcal{V}$, i.e., $e \subseteq \mathcal{V}$. Unlike a graph edge, a hyperedge $e$ may contain more than two nodes. If $\forall e \in \mathcal{E}$ one has $|e| = d \in \mathbb{N}$, the hypergraph $\mathcal{G}$ is termed $d$-uniform. A $d$-uniform hypergraph can be represented by a $d$-dimensional supersymmetric tensor such that for all distinct collections $i_1, \ldots, i_d \in \mathcal{V}$, $\mathbf{A}_{i_1, \ldots, i_d} = \frac{1}{(d-2)!}$ if $e = \{i_1, \ldots, i_d\} \in \mathcal{E}$, and $\mathbf{A}_{i_1, \ldots, i_d} = 0$ otherwise. Henceforth, $\mathbf{A}_{:, \ldots, i_d}$ is used to denote the slice of $\mathbf{A}$ along the first coordinate. A hypergraph can alternatively be represented by its incidence matrix $\mathbf{H}$, where $\mathbf{H}_{ve} = 1$ if $v \in e$ and $\mathbf{H}_{ve} = 0$ otherwise. We use the superscript $(t)$ to represent the functions or variables at the $t$-th step of propagation and $\|$ to denote concatenation. Furthermore, $\boldsymbol{\Theta}$ and $\mathbf{b}$ are reserved for a learnable weight matrix and bias of a neural network, respectively. Finally, we use $\sigma(\cdot)$ to denote a nonlinear activation function (such as ReLU, eLU or LeakyReLU), which depends on the model used.

**CE-based propagation on hypergraphs.** The CE of a hypergraph $\mathcal{G}(\mathcal{V}, \mathcal{E})$ is a weighted graph with the same set of nodes $\mathcal{V}$. It can be described in terms of the associated adjacency or incidence matrices which we write with a slight abuse of notation as $\mathbf{A}_{ij}^{(CE)} = \sum_{i_3, \ldots, i_d \in \mathcal{V}} \mathbf{A}_{i,j,i_3, \ldots, i_d}$ and $\mathbf{H}^{(CE)} = \mathbf{H}\mathbf{H}^T$, respectively. It is obvious that these two matrices only differ in their diagonal entries (0s versus node degrees, respectively). One step of propagation of a $F$-dimensional node feature matrix $\mathbf{X} \in \mathbb{R}^{n \times F}$ is captured by $\mathbf{A}^{(CE)}\mathbf{X}$ or $\mathbf{H}^{(CE)}\mathbf{X}$; alternatively, in terms of node feature updates, we have

$$\text{CEpropA: } \mathbf{X}_{v,:}^{(t+1)} = \sum_{e:v \in e} \sum_{u:u \in e \setminus v} \mathbf{X}_{u,:}^{(t)}; \quad \text{CEpropH: } \mathbf{X}_{v,:}^{(t+1)} = \sum_{e:v \in e} \sum_{u:u \in e} \mathbf{X}_{u,:}^{(t)}, \quad (1)$$

Many existing hypergraph convolutional layers actually perform CE-based propagation, potentially with further degree normalization and nonlinear hyperedge weights. For example, the propagation rule of HGNN (Feng et al., 2019) takes the following node-wise form:

$$\mathbf{X}_{v,:}^{(t+1)} = \sigma \left( \left[ \frac{1}{\sqrt{d_v}} \sum_{e:v \in e} \frac{w_e}{|e|} \sum_{u:u \in e} \frac{\mathbf{X}_{u,:}^{(t)}}{\sqrt{d_u}} \right] \boldsymbol{\Theta}^{(t)} + \mathbf{b}^{(t)} \right), \quad (2)$$

where $d_v$ denotes the degree of node $v$, $w_e$ is a predefined weight of hyperedge $e$ and $\sigma$ is the ReLU activation function. The hypergraph convolution in HCHA (Bai et al., 2021) uses different degree normalizations and attention weights, with the attention weights depending on node features and the hyperedge features. If datasets do not contain hyperedge feature information or if the features come from a different domain compared to the node features, one cannot use their attention module (Bai

et al., 2021). HyperGCN replaces each hyperedge by an incomplete clique via so-called *mediators* (Yadati et al., 2019). When the hypergraph is 3-uniform, the aforementioned approach becomes a standard weighted CE. Hence, all the described hypergraph neural networks adapt propagation rules based on CE or its variants, potentially with the addition of nonlinear hyperedge weights. The described methods achieve reasonable good performance on standard cocitation and coauthor benchmarking datasets.

**Tensor-based propagations.** As mentioned in the introduction, there exist more elaborate tensor-based propagation schemes which in some cases outperform CE-based methods. The propagation rules related to Z eigenproblems such as multilinear PageRank (Gleich et al., 2015) and spacey random walks (Benson et al., 2017) are two such examples. The Z eigenproblem for an adjacency tensor $\mathbf{A}$ of a $d$-uniform hypergraph is defined as:

$$\lambda x = \mathbf{A}x^{d-1} \triangleq \sum_{i_2,\ldots,i_d} \mathbf{A}_{:,i_2,\ldots,i_d} x_{i_2} \ldots x_{i_d}, x \in \mathbb{R}^n \text{ and } \|x\|^2 = 1. \tag{3}$$

Here, $\mathbf{A}x^{d-1}$ equals $\sum_{i_2,\ldots,i_d} \mathbf{A}_{:,i_2,\ldots,i_d} x_{i_2} \ldots x_{i_d}$, an entity frequently used in the tensor analysis literature. The Z eigenproblem has been extensively studied both in the context of tensor analysis and network sciences; the problem is also known as the $l^2$ eigenproblem (Lim, 2005; Gautier et al., 2019). We refer the interested readers to Qi & Luo (2017) for a more detailed theoretical analysis of the Z eigenproblems and Benson (2019) for its application in the study of hypergraph centralities.

By ignoring the norm constraint, one can define the following tensor-based propagation rule based on (3) according to:

$$\text{Zprop: } \mathbf{X}_{v,:}^{(t+1)} = \sum_{e:v\in e} (d-1) \prod_{u:u\in e\setminus v} \mathbf{X}_{u,:}^{(t)}. \tag{4}$$

Despite its interesting theoretical properties, Zprop is known to have what is termed the "unit problem" (Benson, 2019). In practice, the product can cause numerical instabilities for large hyperedges. Furthermore, Zprop has only been studied for the case of $d$-uniform hypergraphs, which makes it less relevant for general hypergraph learning tasks. Clearly, CEprop and Zprop have different advantages and disadvantages for different dataset structures. This motivates finding a general framework that encompasses these two and other propagation rules. In this case, we aim to learn the suitable propagation scheme under such a framework for hypergraph neural networks.

## 3 ALLSET: ONE METHOD TO BIND THEM ALL

We show that all the above described propagation methods can be unified within one setting, termed AllSet. The key observation is that all propagation rules equal a composition of two multiset functions, defined below.

**Definition 3.1.** *A function $f$ is permutation invariant if and only if $\forall \pi \in S_n$, where $S_n$ denotes the symmetric group of order $n!$, $f(\mathbf{x}_{\pi(1)}, \ldots, \mathbf{x}_{\pi(n)}) = f(\mathbf{x}_1, \ldots, \mathbf{x}_n)$.*

**Definition 3.2.** *We say that a function $f$ is a multiset function if it is permutation invariant.*

Next, let $V_{e,\mathbf{X}} = \{\mathbf{X}_{u,:} : u \in e\}$ denote the multiset of hidden node representations contained in the hyperedge $e$. Also, let $\mathbf{Z} \in \mathbb{R}^{|\mathcal{E}| \times F'}$ denote the hidden hyperedge representations. Similarly, let $E_{v,\mathbf{Z}} = \{\mathbf{Z}_{e,:} : v \in e\}$ be the multiset of hidden representations of hyperedges that contain the node $v$. The AllSet framework uses the update rules

$$\mathbf{Z}_{e,:}^{(t+1)} = f_{\mathcal{V}\to\mathcal{E}}(V_{e,\mathbf{X}^{(t)}}; \mathbf{Z}_{e,:}^{(t)}), \quad \mathbf{X}_{v,:}^{(t+1)} = f_{\mathcal{E}\to\mathcal{V}}(E_{v,\mathbf{Z}^{(t+1)}}; \mathbf{X}_{v,:}^{(t)}), \tag{5}$$

where $f_{\mathcal{V}\to\mathcal{E}}$ and $f_{\mathcal{E}\to\mathcal{V}}$ are two multiset functions with respect to their first input. For the initial condition, we choose $\mathbf{Z}_{e,:}^{(0)}$ and $\mathbf{X}_{v,:}^{(0)}$ to be hyperedge features and node features, respectively (if available). If these are not available, we set both entities to be all-zero matrices. Note that we make the tacit assumption that both functions $f_{\mathcal{V}\to\mathcal{E}}$ and $f_{\mathcal{E}\to\mathcal{V}}$ also include the hypergraph $\mathcal{G}$ as an input. This allows degree normalization to be part of our framework. As one can also distinguish the aggregating node $v$ from the multiset $V_{e,\mathbf{X}^{(t)}}$, we also have the following AllSet variant:

$$\mathbf{Z}_{e,:}^{(t+1),v} = f_{\mathcal{V}\to\mathcal{E}}(V_{e\setminus v,\mathbf{X}^{(t)}}; \mathbf{Z}_{e,:}^{(t),v}, \mathbf{X}_{v,:}^{(t)}), \quad \mathbf{X}_{v,:}^{(t+1)} = f_{\mathcal{E}\to\mathcal{V}}(E_{v,\mathbf{Z}^{(t+1),v}}; \mathbf{X}_{v,:}^{(t)}), \tag{6}$$

For simplicity, we omit the last input argument $\mathbf{X}_{v,:}^{(t)}$ for $f_{\mathcal{V} \to \mathcal{E}}$ in (6), unless explicitly needed (as in the proof pertaining to HyperGCN). The formulation (5) lends itself to a significantly more computationally- and memory-efficient pipeline. This is clearly the case since for each hyperedge $e$, the expression in (5) only uses one hyperedge hidden representation while the expression (6) uses $|e|$ distinct hidden representations. This difference can be substantial when the hyperedge sizes are large. Hence, we only use (6) for theoretical analysis purposes and (5) for all experimental verifications. The next theorems establish the universality of the AllSet framework.

**Theorem 3.3.** *CEpropH of* (1) *is a special case of AllSet* (5). *Furthermore, CEpropA of* (1) *and Zprop of* (4) *are also special cases of AllSet* (6).

*Sketch of proof.* If we ignore the second input and choose both $f_{\mathcal{V} \to \mathcal{E}}$ and $f_{\mathcal{E} \to \mathcal{V}}$ to be sums over their input multisets, we recover the CE-based propagation rule CEpropH of (1) via (5). For CEpropA, the same observation is true with respect to (6). If one instead chooses $f_{\mathcal{V} \to \mathcal{E}}$ to be the product over its input multiset, multiplied by the scalar $|V_{e,\mathbf{X}^{(t)}}| - 1$, one recovers the tensor-based propagation Zprop of (4) via (6).

Next, we show that many state-of-the-art hypergraph neural network layers also represent special instances of AllSet and are strictly less expressive than AllSet.

**Theorem 3.4.** *The hypergraph neural network layers of HGNN* (2)*, HCHA (Bai et al., 2021), HNHN (Dong et al., 2020) and HyperSAGE (Arya et al., 2020) are all special cases of AllSet* (5). *The HyperGCN layer (Yadati et al., 2019) is a special instance of AllSet* (6). *Furthermore, all the above methods are strictly less expressive than AllSet* (5) *and* (6). *More precisely, there exists a combination of multiset functions $f_{\mathcal{V} \to \mathcal{E}}$ and $f_{\mathcal{E} \to \mathcal{V}}$ for AllSet* (5) *and* (6) *that cannot be modeled by any of the aforementioned hypergraph neural network layers.*

The first half of the proof is by direct construction. The second half of the proof consists of counter-examples. Intuitively, none of the listed hypergraph neural network layers can model Zprop (3) while we established in the previous result that Zprop is a special case of AllSet.

The last result shows that the MPNN (Gilmer et al., 2017) framework is also a special instance of our AllSet for graphs, which are (clearly) special cases of hypergraphs. Hence, AllSet may also be viewed as a hypergraph generalization of MPNN. Note that MPNN itself generalizes many well-known GNNs, such as GCN (Kipf & Welling, 2017), Gated Graph Neural Networks (Li et al., 2015) and GAT (Veličković et al., 2018).

**Theorem 3.5.** *MPNN is a special case of AllSet* (6) *when applied to graphs.*

## 4 How to Learn AllSet Layers

The key idea behind AllSet is to *learn* the multiset functions $f_{\mathcal{V} \to \mathcal{E}}$ and $f_{\mathcal{E} \to \mathcal{V}}$ on the fly for each dataset and task. To facilitate this learning process, we first have to properly parametrize the multiset functions. Ideally, the parametrization should represent a universal approximator for a multiset function that allows one to retain the higher expressive power of our architectures when compared to that of the hypergraph neural networks described in Theorem 3.4. For simplicity, we focus on the multiset inputs of $f_{\mathcal{V} \to \mathcal{E}}$ and $f_{\mathcal{E} \to \mathcal{V}}$ and postpone the discussion pertaining to the second arguments of the functions to the end of this section.

Under the assumption that the multiset size is finite, the authors of Zaheer et al. (2017) and Wagstaff et al. (2019) proved that any multiset functions $f$ can be parametrized as $f(S) = \rho \left( \sum_{s \in S} \phi(s) \right)$, where $\rho$ and $\phi$ are some bijective mappings (Theorem 4.4 in Wagstaff et al. (2019)). In practice, these mappings can be replaced by any universal approximator such as a multilayer perceptron (MLP) (Zaheer et al., 2017). This leads to the purely MLP AllSet layer for hypergraph neural networks, termed AllDeepSets.

$$\text{AllDeepSets: } f_{\mathcal{V} \to \mathcal{E}}(S) = f_{\mathcal{E} \to \mathcal{V}}(S) = \text{MLP} \left( \sum_{s \in S} \text{MLP}(s) \right). \tag{7}$$

The authors of Lee et al. (2019) argued that the unweighted sum in Deep Set makes it hard to learn the importance of each individual contributing term. Thus, they proposed the Set Transformer

paradigm which was shown to offer better performance than Deep Sets as a learnable multiset function architecture. Based on this result, we also propose an attention-based AllSet layer for hypergraph neural networks, termed AllSetTransformer. Given the matrix $\mathbf{S} \in \mathbb{R}^{|S| \times F}$ which represents the multiset $S$ of $F$-dimensional real vectors, the definition of AllSetTransformer is

$$\text{AllSetTransformer: } f_{\mathcal{V} \to \mathcal{E}}(S) = f_{\mathcal{E} \to \mathcal{V}}(S) = \text{LN}\left(\mathbf{Y} + \text{MLP}(\mathbf{Y})\right),$$

$$\text{where } \mathbf{Y} = \text{LN}\left(\theta + \text{MH}_{h,\omega}(\theta, \mathbf{S}, \mathbf{S})\right), \text{MH}_{h,\omega}(\theta, \mathbf{S}, \mathbf{S}) = \|_{i=1}^{h} \mathbf{O}^{(i)},$$

$$\mathbf{O}^{(i)} = \omega\left(\theta^{(i)}(\mathbf{K}^{(i)})^T\right) \mathbf{V}^{(i)}, \theta \triangleq \|_{i=1}^{h} \theta^{(i)}, \ \mathbf{K}^{(i)} = \text{MLP}^{K,i}(\mathbf{S}), \ \mathbf{V}^{(i)} = \text{MLP}^{V,i}(\mathbf{S}). \quad (8)$$

Here, LN represents the layer normalization (Ba et al., 2016), $\|$ denotes concatenation and $\theta \in \mathbb{R}^{1 \times hF_h}$ is a learnable weight; in addition, $\text{MH}_{h,\omega}$ is a multihead attention mechanism with $h$ heads and activation function $\omega$ (Vaswani et al., 2017). Note that the dimension of the output of $\text{MH}_{h,\omega}$ is $1 \times hF_h$ where $F_h$ is the hidden dimension of $\mathbf{V}^{(i)} \in \mathbb{R}^{|S| \times F_h}$. In our experimental setting, we choose $\omega$ to be the softmax function which is robust in practice. Our formulation of projections in $\text{MH}_{h,\omega}$ is slight more general than standard linear projections. If one restricts $\text{MLP}^{K,i}$ and $\text{MLP}^{V,i}$ to one-layer perceptrons without a bias term (i.e., including only a learnable weight matrix), we obtain standard linear projections in the multihead attention mechanism. The output dimensions of $\text{MLP}^{K,i}$ and $\text{MLP}^{V,i}$ are both equal to $\mathbb{R}^{|S| \times F_h}$. It is worth pointing out that all the MLP modules operate row-wise, which means they are applied to each multiset element (a real vector) independently and in an identical fashion. This directly implies that MLP modules are permutation equivariant.

Based on the above discussion and Proposition 2 of Lee et al. (2019) we have the following result.

**Proposition 4.1.** *The functions $f_{\mathcal{V} \to \mathcal{E}}$ and $f_{\mathcal{E} \to \mathcal{V}}$ defined in AllSetTransformer* (8) *are permutation invariant. Furthermore, the functions $f_{\mathcal{V} \to \mathcal{E}}$ and $f_{\mathcal{E} \to \mathcal{V}}$ defined in* (8) *are universal approximators of multiset functions when the size of the input multiset is finite.*

Note that in practice, both the size of hyperedges and the node degrees are finite. Thus, the finite size multiset assumption is satisfied. Therefore, the above results directly imply that the AllDeepSets (7) and AllSetTransformer (8) layers have the same expressive power as the general AllSet framework. Together with the result of Theorem 3.4, we arrive at the conclusion that AllDeepSets and AllSetTransformer are both more expressive than any other aforementioned hypergraph neural network.

Conceptually, one can also incorporate the Set Attention Block and the two steps pooling strategy of Lee et al. (2019) into AllSet. However, it appears hard to make such an implementation efficient and we hence leave this investigation as future work. In practice, we find that even our simplified design (8) already outperforms all baseline hypergraph neural networks, as demonstrated by our extensive experiments. Also, it is possible to utilize the information of the second argument of $f_{\mathcal{V} \to \mathcal{E}}$ and $f_{\mathcal{E} \to \mathcal{V}}$ via concatenation, another topic relegated to future work.

As a concluding remark, we observe that the multiset functions in both AllDeepSets (7) and AllSetTransformer (8) are universal approximators for general multiset functions. In contrast, the other described hypergraph neural network layers fail to retain the universal approximation property for general multiset functions, as shown in Theorem 3.4. This implies that all these hypergraph neural network layers have strictly weaker expressive power compared to AllDeepSets (7) and AllSetTransformer (8). We also note that any other universal approximators for multiset functions can easily be combined with our AllSet framework (as we already demonstrated by our Deep Sets and Set Transformer examples). One possible candidate is Janossy pooling (Murphy et al., 2019) which will be examined as part of our future work. Note that more expressive models do not necessarily have better performance than less expressive models, as many other factors influence system performance (as an example, AllDeepSet performs worse than AllSetTransformer albeit both have the same theoretical expressive power; this is in agreement with the observation from Lee et al. (2019) that Set Transformer can learn multiset functions better than Deep Set.). Nevertheless, we demonstrate in the experimental verification section that our AllSetTransformer indeed has consistently better performance compared to other baseline methods.

## 5 RELATED WORKS

Due to space limitations, a more comprehensive discussion of related work is relegated to the Appendix A.

**Hypergraph learning.** Learning on hypergraphs has attracted significant attention due to its ability to capture higher-order structural information (Li et al., 2017; 2019). In the statistical learning and information theory literature, researchers have studied community detection problem for hypergraph stochastic block models (Ghoshdastidar & Dukkipati, 2015; Chien et al., 2018; 2019a). In the machine learning community, methods that mitigate drawbacks of CE have been reported in (Li & Milenkovic, 2017; Chien et al., 2019b; Hein et al., 2013; Chan et al., 2018). Despite these advances, hypergraph neural network developments are still limited beyond the works covered by the AllSet paradigm. Exceptions include Hyper-SAGNN (Zhang et al., 2019), LEGCN (Yang et al., 2020) and MPNN-R (Yadati, 2020), the only three hypergraph neural networks that do not represent obvious special instances of AllSet. But, Hyper-SAGNN focuses on hyperedge prediction while we focus on semi-supervised node classification. LEGCN first transforms hypergraphs into graphs via line expansion and then applies a standard GCN. Although the transformation used in LEGCN does not cause a large distortion as CE, it significantly increases the number of nodes and edges. Hence, it is not memory and computationally efficient (more details about this approach can be found in the Appendix G). MPNN-R focuses on recursive hypergraphs (Joslyn & Nowak, 2017), which is not the topic of this work. It is an interesting future work to investigate the relationship between AllSet and Hyper-SAGNN or MPNN-R when extending AllSet to hyperedge prediction tasks or recursive hypergraph implementations.

Furthermore, Tudisco & Higham (2021); Tudisco et al. (2021) also proposed to develop and analyze hypergraph propagations that operate "between" CE-prop and Z-prop rules. These works focus on the theoretical analysis of special types of propagation rules, which can also be modeled by our AllSet frameworks. Importantly, our AllSet layers do not fix the rules but instead learn them in an adaptive manner. Also, one can view Tudisco & Higham (2021); Tudisco et al. (2021) as the hypergraph version of SGC, which can help with interpreting the functionalities of AllSet layers.

**Star expansion, UniGNN and Heterogeneous GNNs.** A very recent work concurrent to ours, UniGNN (Huang & Yang, 2021), also proposes to unify hypergraph and GNN models. Both Huang & Yang (2021) and our work can be related to hypergraph star expansion (Agarwal et al., 2006; Yang et al., 2020), which results in a bipartite graph (see Figure 1). One particular variant, Uni-GIN, is related to our AllDeepSets model, as both represent a hypergraph generalization of GIN (Xu et al., 2019). However, UniGNN does not make use of deep learning methods for learning multiset functions, which is crucial for identifying the most appropriate propagation and aggregation rules for individual datasets and learning tasks as done by AllSetTransformer (see Section 6 for supporting information). Also, the most advanced variant of UniGNN, UniGCNII, is not comparable to AllSetTransformers. Furthermore, one could also naively try to apply heterogeneous GNNs, such as HAN (Wang et al., 2019b), to hypergraph datasets converted into bipartite graphs. This approach was not previously examined in the literature, but we experimentally tested it nevertheless to show that the performance improvements in this case are significantly smaller than ours and that the method does not scale well even for moderately sized hypergraphs. Another very recent work (Xue et al., 2021) appears at first glance similar to ours, but it solves a very different problem by transforming a *heterogeneous bipartite graph* into a hypergraph and then apply standard GCN layers to implement $f_{\mathcal{V} \rightarrow \mathcal{E}}$ and $f_{\mathcal{E} \rightarrow \mathcal{V}}$, which is more related to HNHN (Dong et al., 2020) and UniGCN (Huang & Yang, 2021) and very different from AllSetTransformer.

**Learning (multi)set functions.** (Multi)set functions, which are also known as pooling architectures for (multi)sets, have been used in numerous problems including causality discovery (Lopez-Paz et al., 2017), few-shot image classification (Snell et al., 2017) and conditional regression and classification (Garnelo et al., 2018). The authors of Zaheer et al. (2017) provide a universal way to parameterize the (multi)set functions under some mild assumptions. Similar results have been proposed independently by the authors of Qi et al. (2017) for computer vision applications. The authors of Lee et al. (2019) propose to learn multiset functions via attention mechanisms. This further inspired the work in Baek et al. (2021), which adopted the idea for graph representation learning. The authors of Wagstaff et al. (2019) discuss the limitation of Zaheer et al. (2017), and specifically focus on continuous functions. A more general treatment of how to capture complex dependencies among multiset elements is available in Murphy et al. (2019). To the best of our knowledge, this work is the first to build the connection between learning multiset functions with propagations on hypergraph.

Table 1: Dataset statistics: $|e|$ refers to the size of the hyperedges while $d_v$ refers to the node degree.

|  | Cora | Citeseer | Pubmed | Cora-CA | DBLP-CA | Zoo | 20News | Mushroom | NTU2012 | ModelNet40 | Yelp | House | Walmart |
|---|---|---|---|---|---|---|---|---|---|---|---|---|---|
| $|\mathcal{V}|$ | 2708 | 3312 | 19717 | 2708 | 41302 | 101 | 16242 | 8124 | 2012 | 12311 | 50758 | 1290 | 88860 |
| $|\mathcal{E}|$ | 1579 | 1079 | 7963 | 1072 | 22363 | 43 | 100 | 298 | 2012 | 12311 | 679302 | 341 | 69906 |
| # feature | 1433 | 3703 | 500 | 1433 | 1425 | 16 | 100 | 22 | 100 | 100 | 1862 | 100 | 100 |
| # class | 7 | 6 | 3 | 7 | 6 | 7 | 4 | 2 | 67 | 40 | 9 | 2 | 11 |
| max $|e|$ | 5 | 26 | 171 | 43 | 202 | 93 | 2241 | 1808 | 5 | 5 | 2838 | 81 | 25 |

## 6 EXPERIMENTS

We focus on semi-supervised node classification in the transductive setting. We randomly split the data into training/validation/test samples using $(50\%/25\%/25\%)$ splitting percentages. We aggregate the results of 20 experiments using multiple random splits and initializations.

**Methods used for comparative studies.** We compare our AllSetTransformer and AllDeepSets with ten baseline models, MLP, CE+GCN, CE+GAT, HGNN (Feng et al., 2019), HCHA (Bai et al., 2021), HyperGCN (Yadati et al., 2019), HNHN (Dong et al., 2020), UniGCNII (Huang & Yang, 2021) and HAN (Wang et al., 2019b), with both full batch and mini-batch settings. All architectures are implemented using the Pytorch Geometric library (PyG) (Fey & Lenssen, 2019) except for HAN, in which case the implementation was retrieved from Deep Graph Library (DGL) (Wang et al., 2019a). The implementation of HyperGCN[2] is used in the same form as reported in the official repository. We adapted the implementation of HCHA from PyG. We also reimplemented HNHN and HGNN using the PyG framework. Note that in the original implementation of HGNN[3], propagation is performed via matrix multiplication which is far less memory and computationally efficient when compared to our implementation. MLP, GCN and GAT are executed directly from PyG. The UniGCNII code is taken from the official site[4]. For HAN, we tested both the full batch training setting and a mini-batch setting provided by DGL. We treated the vertices ($\mathcal{V}$) and hyperedges ($\mathcal{E}$) as two heterogeneous types of nodes, and used two metapaths: $\{\mathcal{V} \rightarrow \mathcal{E} \rightarrow \mathcal{V}, \mathcal{E} \rightarrow \mathcal{V} \rightarrow \mathcal{E}\}$ in its semantic attention. In the mini-batch setting, we used the DGL's default neighborhood sampler with a number of neighbors set to 32 during training and 64 for validation and testing purposes. In both settings, due to the specific conversion of hyperedges into nodes, our evaluations only involve the original labeled vertices in the hypergraph and ignore the hyperedge node-proxies. More details regarding the experimental settings and the choices of hyperparameters are given in the Appendix J.

**Benchmarking and new datasets.** We use ten available datasets from the existing hypergraph neural networks literature and three newly curated datasets from different application domains. The benchmark datasets include cocitation networks Cora, Citeseer and Pubmed, downloaded from Yadati et al. (2019). The coauthorship networks Cora-CA and DBLP-CA were adapted from Yadati et al. (2019). We also tested datasets from the UCI Categorical Machine Learning Repository (Dua & Graff, 2017), including 20Newsgroups, Mushroom and Zoo. Datasets from the area of computer vision and computer graphics include ModelNet40 (Wu et al., 2015) and NTU2012 (Chen et al., 2003). The hypergraph construction follows the recommendations from Feng et al. (2019); Yang et al. (2020). The three newly introduced datasets are adaptations of Yelp (Yelp), House (Chodrow et al., 2021) and Walmart (Amburg et al., 2020). Since there are no node features in the original datasets of House and Walmart, we use Gaussian random vectors instead, in a fashion similar to what was proposed for Contextual stochastic block models (Deshpande et al., 2018). We use postfix $(x)$ to indicate the standard deviation of the added Gaussian features. The partial statistics of all datasets are provided in Table 1, and more details are available in the Appendix I.

**Results.** We use accuracy (micro-F1 score) as the evaluation metric, along with its standard deviation. The relevant findings are summarized in Table 2. The results show that AllSetTransformer is the most robust hypergraph neural network model and that it has the best overall performance when compared to state-of-the art and new models such a hypergraph HAN. In contrast, all baseline methods perform poorly on at least two datasets. For example, UniGCNII and HAN are two of the best performing baseline models according to our experiments. However, UniGCNII performs poorly on Zoo, Yelp and Walmart when compared to our AllSetTransformer. More precisely, AllSetTransformer outperforms UniGCNII by $11.01\%$ in accuracy on the Walmart(1) dataset. HAN has poor performance when compared AllSetTransformer on Mushroom, NTU2012 and ModelNet40. Fur-

---

[2] https://github.com/malllabiisc/HyperGCN    [3] https://github.com/iMoonLab/HGNN    [4] https://github.com/OneForward/UniGNN

Table 2: Results for the tested datasets: Mean accuracy (%) ± standard deviation. Boldfaced letters shaded grey are used to indicate the best result, while blue shaded boxes indicate results within one standard deviation of the best result. NA indicates that the method has numerical precision issue. For HAN*, additional preprocessing of each dataset is required (see the Section 6 for more details).

| | Cora | Citeseer | Pubmed | Cora-CA | DBLP-CA | Zoo | 20Newsgroups | mushroom |
|---|---|---|---|---|---|---|---|---|
| AllSetTransformer | 78.59 ± 1.47 | 73.08 ± 1.20 | 88.72 ± 0.37 | 83.63 ± 1.47 | 91.53 ± 0.23 | 97.50 ± 3.59 | 81.38 ± 0.58 | 100.00 ± 0.00 |
| AllDeepSets | 76.88 ± 1.80 | 70.83 ± 1.63 | 88.75 ± 0.33 | 81.97 ± 1.50 | 91.27 ± 0.27 | 95.39 ± 4.77 | 81.06 ± 0.54 | 99.99 ± 0.02 |
| MLP | 75.17 ± 1.21 | 72.67 ± 1.56 | 87.47 ± 0.51 | 74.31 ± 1.89 | 84.83 ± 0.22 | 87.18 ± 4.44 | 81.42 ± 0.49 | 100.00 ± 0.00 |
| CECGN | 76.17 ± 1.39 | 70.16 ± 1.31 | 86.45 ± 0.43 | 77.05 ± 1.26 | 88.00 ± 0.26 | 51.54 ± 11.19 | OOM | 95.27 ± 0.47 |
| CEGAT | 76.41 ± 1.53 | 70.63 ± 1.30 | 86.81 ± 0.42 | 76.16 ± 1.19 | 88.59 ± 0.29 | 47.88 ± 14.03 | OOM | 96.60 ± 1.67 |
| HNHN | 76.36 ± 1.92 | 72.64 ± 1.57 | 86.90 ± 0.30 | 77.19 ± 1.49 | 86.78 ± 0.29 | 93.59 ± 5.88 | 81.35 ± 0.61 | 100.00 ± 0.01 |
| HGNN | 79.39 ± 1.36 | 72.45 ± 1.16 | 86.44 ± 0.44 | 82.64 ± 1.65 | 91.03 ± 0.20 | 92.50 ± 4.58 | 80.33 ± 0.42 | 98.73 ± 0.32 |
| HCHA | 79.14 ± 1.02 | 72.42 ± 1.42 | 86.41 ± 0.36 | 82.55 ± 0.97 | 90.92 ± 0.22 | 93.65 ± 6.15 | 80.33 ± 0.80 | 98.70 ± 0.39 |
| HyperGCN | 78.45 ± 1.26 | 71.28 ± 0.82 | 82.84 ± 8.67 | 79.48 ± 2.08 | 89.38 ± 0.25 | N/A | 81.05 ± 0.59 | 47.90 ± 1.04 |
| UniGCNII | 78.81 ± 1.05 | 73.05 ± 2.21 | 88.25 ± 0.40 | 83.60 ± 1.14 | 91.69 ± 0.19 | 93.65 ± 4.37 | 81.12 ± 0.67 | 99.96 ± 0.05 |
| HAN (full batch)* | 80.18 ± 1.15 | 74.05 ± 1.43 | 86.21 ± 0.48 | 84.04 ± 1.02 | 90.89 ± 0.23 | 85.19 ± 8.18 | OOM | 90.86 ± 2.40 |
| HAN (mini batch)* | 79.70 ± 1.77 | 74.12 ± 1.52 | 85.32 ± 2.25 | 81.71 ± 1.73 | 90.17 ± 0.65 | 75.77 ± 7.10 | 79.72 ± 0.62 | 93.45 ± 1.31 |

| | NTU2012 | ModelNet40 | Yelp | House(1) | Walmart(1) | House(0.6) | Walmart(0.6) | avg. ranking (↑) |
|---|---|---|---|---|---|---|---|---|
| AllSetTransformer | 88.69 ± 1.24 | 98.20 ± 0.20 | 36.89 ± 0.51 | 69.33 ± 2.20 | 65.46 ± 0.25 | 83.14 ± 1.92 | 78.46 ± 0.40 | 2.00 |
| AllDeepSets | 88.09 ± 1.52 | 96.98 ± 0.26 | 30.36 ± 1.57 | 67.82 ± 2.40 | 64.55 ± 0.33 | 80.70 ± 1.59 | 78.46 ± 0.26 | 4.47 |
| MLP | 85.52 ± 1.49 | 96.14 ± 0.36 | 31.96 ± 0.44 | 67.93 ± 2.33 | 45.51 ± 0.24 | 81.53 ± 2.26 | 63.28 ± 0.37 | 6.27 |
| CECGN | 81.52 ± 1.43 | 89.92 ± 0.46 | OOM | 62.80 ± 2.61 | 54.44 ± 0.24 | 64.36 ± 2.41 | 59.78 ± 0.32 | 9.66 |
| CEGAT | 82.21 ± 1.23 | 92.52 ± 0.39 | OOM | 69.09 ± 3.00 | 51.14 ± 0.56 | 77.25 ± 2.53 | 59.47 ± 1.05 | 8.80 |
| HNHN | 89.11 ± 1.44 | 97.84 ± 0.25 | 31.65 ± 0.44 | 67.80 ± 2.59 | 47.18 ± 0.35 | 78.78 ± 1.88 | 65.80 ± 0.39 | 5.87 |
| HGNN | 87.72 ± 1.35 | 95.44 ± 0.33 | 33.04 ± 0.62 | 61.39 ± 2.96 | 62.00 ± 0.24 | 66.16 ± 1.80 | 77.72 ± 0.21 | 5.73 |
| HCHA | 87.48 ± 1.87 | 94.48 ± 0.28 | 30.99 ± 0.72 | 61.36 ± 2.53 | 62.45 ± 0.26 | 67.91 ± 2.26 | 77.12 ± 0.26 | 6.40 |
| HyperGCN | 56.36 ± 4.86 | 75.89 ± 5.26 | 29.42 ± 1.54 | 48.31 ± 2.93 | 44.74 ± 2.81 | 78.22 ± 2.46 | 55.31 ± 0.30 | 9.87 |
| UniGCNII | 89.30 ± 1.33 | 98.07 ± 0.23 | 31.70 ± 0.52 | 67.25 ± 2.57 | 54.45 ± 0.37 | 80.65 ± 1.96 | 72.08 ± 0.28 | 3.87 |
| HAN (full batch)* | 83.58 ± 1.46 | 94.04 ± 0.41 | OOM | 71.05 ± 2.26 | OOM | 83.27 ± 1.62 | OOM | 6.73 |
| HAN (mini batch)* | 80.77 ± 2.36 | 91.52 ± 0.96 | 26.05 ± 1.37 | 62.00 ± 9.06 | 48.57 ± 1.04 | 82.04 ± 2.68 | 63.10 ± 0.96 | 7.60 |

thermore, it experiences memory overload issues on Yelp and Walmart with a full batch setting and performs poorly with a mini-batch setting on the same datasets. Numerically, AllSetTransformer outperforms HAN with full batch setting by $9.14\%$ on the Mushroom and HAN with mini-batch setting and by $16.89\%$ on Walmart(1). These findings emphasize the importance of including diverse datasets from different application domains when trying to test hypergraph neural networks fairly. Testing standard benchmark datasets such as Cora, Citeseer and Pubmed alone is insufficient and can lead to biased conclusions. Indeed, our experiments show that *all* hypergraph neural networks work reasonably well on these three datasets. Our results also demonstrate the power of Set Transformer when applied to hypergraph learning via the AllSet framework.

The execution times of all methods are available in the Appendix J: They show that the complexity of AllSet is comparable to that of the baseline hypergraph neural networks. This further strengthens the case for AllSet, as its performance gains do not come at the cost of high computational complexity. Note that in HAN's mini-batch setting, the neighborhood sampler is performed on CPU instead of GPU, hence its training time is significantly higher than that needed by other methods, due to the frequent I/O operation between CPU and GPU. On some larger datasets such as Yelp and Walmart, 20 runs take more than 24 hours so we only recorded the results for first 10 runs.

The performance of AllDeepSets is not on par with that of AllSetTransformer, despite the fact that both represent universal approximators of the general AllSet formalism. This result confirms the assessment of Lee et al. (2019) that attention mechanisms are crucial for learning multiset functions in practice. Furthermore, we note that combining CE with existing GNNs such as GCN and GAT leads to suboptimal performance. CE-based approaches are also problematic in terms of memory efficiency when large hyperedges are present. This is due to the fact the CE of a hyperedge $e$ leads to $\Theta(|e|^2)$ edges in the resulting graph. Note that as indicated in Table 2, CE-based methods went out-of-memory (OOM) for 20Newsgroups and Yelp. These two datasets have the largest maximum hyperedge size, as can be see from Table 1. HAN encounters the OOM issue on even more datasets when used in the full batch setting, and its mini-batch setting mode perform poorly on Yelp and Walmart. This shows that a naive application of standard heterogeneous GNNs on large hypergraphs often fails and is thus not as robust as our AllSetTransformer. The mediator HyperGCN approach exhibits numerical instabilities on some datasets (i.e., Zoo and House) and may hence not be suitable for learning on general hypergraph datasets.

ACKNOWLEDGMENTS

The authors would like to thank Prof. Pan Li at Purdue University for helpful discussions. The authors would also like to thank Chaoqi Yang for answering questions regarding the LEGCN method. This work was funded by the NSF grant 1956384.

## 7 ETHICS STATEMENT

We do not aware of any potential ethical issues regarding our work. For the newly curated three datasets, there is no private personally identifiable information. Note that although nodes in House datasets represent congresspersons and hyperedges represent members serving on the same committee, this information is public and not subject to privacy constraints. For the Walmart dataset, the nodes are products and hyperedges are formed by sets of products purchased together. We do not include any personal information regarding the buyers (customers) in the text. In the Yelp dataset the nodes represent restaurants and hyperedges are collections of restaurants visited by the same user. Similar to the Walmart case, there is no user information included in the dataset. More details of how the features in each datasets are generated can be found in Appendix I.

## 8 REPRODUCIBILITY STATEMENT

We have tried our best to ensure reproducibility of our results. We built a succinct pipeline for standardization of the hypergraph neural networks evaluation process based on Pytorch Geometric. This implementation can be checked by referring to our supplementary material. We include all dataset in our supplementary material and integrate all tested methods in our code. Hence, one can simply reproduce all experiment results effortlessly (by just running a one-line command). All details regarding how the datasets were prepared are stated in Appendix I. All choices of hyperparameters and the process of selecting them are available in Appendix J. Our experimental settings including the specifications of our machine and environment, the training/validation/test split and the evaluation metric. Once again, refer ro the Appendix J and Section 6. We also clearly specify all the package dependencies used in our code in the Supplementary material.

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

APPENDIX

## A  ADDITIONAL RELATED WORKS

We further discuss related works for completeness.

**Graph neural networks on graphs.** GNNs have received significant attention in the past, often under the umbrella of Geometric Deep Learning (Bronstein et al., 2017). Many successful architectures have been proposed for various tasks on graphs, including GCN (Kipf & Welling, 2017), GAT (Vaswani et al., 2017), GraphSAGE (Hamilton et al., 2017), GIN (Xu et al., 2019) and many others (Klicpera et al., 2019; Wu et al., 2019; Velickovic et al., 2019; Li et al., 2020; Xu et al., 2018; Chien et al., 2021b). There have also been some recent attempts to use Transformers in GNNs (Baek et al., 2021; Yun et al., 2019; Hu et al., 2020b). Nevertheless, all these methods only apply to graphs and not to hypergraphs. Although CE can be used in all these case, this is clearly not an optimal strategy due to the previously mentioned distortion issues. There are also many techniques that can improve GNNs in various directions. For example, PairNorm (Zhao & Akoglu, 2019) and DropEdge (Rong et al., 2019) allow one to build deeper GNNs. ClusterGCN (Chiang et al., 2019) and GraphSAINT (Zeng et al., 2019), on the other hand, may be used to significantly scale up GNNs. Residual correlation (Jia & Benson, 2020) and CopulaGNN (Ma et al., 2021) have been shown to improve GNNs on graph regression problems. These works highlight new directions and limitations of not only our method, but also hypergraph neural networks in general. Nevertheless, we believe that AllSet layers can be adapted to resolve these issues using similar ideas as those proposed for graph layers.

**Additional contributions to learning on hypergraphs.** After the submission of our manuscript, we were made aware of some loosely related lines of works. Ding et al. (2020) and Wang et al. (2021) proposed hypergraph attention networks based on the $\mathcal{V} \rightarrow \mathcal{E}$ and $\mathcal{E} \rightarrow \mathcal{V}$ formulation, specialized for certain downstream tasks. However, these approaches fall under our AllSet framework and have smaller expressiveness compared to AllSet. The work Jo et al. (2021) proposes to learn edge representations in a *graph* via message passing on its dual hypergraph. Although the authors used GMT (Baek et al., 2021) as their node pooling module, they do not explore the idea of treating $f_{\mathcal{V} \rightarrow \mathcal{E}}$ and $f_{\mathcal{E} \rightarrow \mathcal{V}}$ as multiset functions for hypergraph neural networks as AllSet does. It is also possible to define propagation schemes based on other tensor eigenproblems, such as H eigenproblems (Qi & Luo, 2017). Defining propagation rules based on H eigenproblems can lead to imaginary features which is problematic: A more detailed discussion regarding this direction is available in the Appendix H.

## B  CONCLUSIONS

We proposed AllSet, a novel hypergraph neural network paradigm that represents a highly general framework for hypergraph neural networks. We implemented hypergraph neural network layers as compositions of two multiset functions that can be efficiently learned for each task and each dataset. Furthermore, for the first time, we integrated Deep Set and Set Transformer methods within hypergraph neural networks for the purpose of learning the described multiset functions. Our theoretical analysis demonstrated that most of the previous hypergraph neural networks are strictly less expressive then our proposed AllSet framework. We conducted the most extensive experiments to date involving ten known benchmarking datasets and three newly curated datasets that represent significant challenges for hypergraph node classification. The results showed that our method has the unique ability to either match or outperform all other baselines, including the state-of-the-art hypergraph neural networks UniGNN (Huang & Yang, 2021). Remarkably, our baseline also included HAN (Wang et al., 2019b), the heterogeneous hypergraph neural networks, via star expansion as the new comparative method in the hypergraph neural network literature for the first time. Our results show that leveraging powerful multiset function learners in our AllSet framework is indeed helpful for hypergraph learning. As a promising future direction, more advanced multiset function learners such as Janossy pooling (Murphy et al., 2019) can be used in our AllSet framework and potentially further improve the performance. We left this topic in future works.

## C    PROOF OF THEOREM 3.3

First, we show that one can recover CEpropH (1) from AllSet (5). By choosing $f_{\mathcal{V}\to\mathcal{E}}$ and $f_{\mathcal{E}\to\mathcal{V}}$ to be the sums over appropriate input multisets and by ignoring the second input, we obtain

$$\mathbf{Z}_{e,:}^{(t+1)} = f_{\mathcal{V}\to\mathcal{E}}(V_{e,\mathbf{X}^{(t)}}; \mathbf{Z}_{e,:}^{(t)}) = \sum_{x\in V_{e,\mathbf{X}^{(t)}}} x = \sum_{u\in e} \mathbf{X}_{u,:}^{(t)}, \qquad (9)$$

$$\mathbf{X}_{v,:}^{(t+1)} = f_{\mathcal{E}\to\mathcal{V}}(E_{v,\mathbf{Z}^{(t+1)}}; \mathbf{X}_{v,:}^{(t)}) = \sum_{x\in E_{v,\mathbf{Z}^{(t+1)}}} x = \sum_{e:v\in e} \mathbf{Z}_{e,:}^{(t+1)}. \qquad (10)$$

Combining these two equations gives

$$\mathbf{X}_{v,:}^{(t+1)} = \sum_{e:v\in e} \mathbf{Z}_{e,:}^{(t+1)} = \sum_{e:v\in e}\sum_{u\in e} \mathbf{X}_{u,:}^{(t)}, \qquad (11)$$

which matches the update equations of CEpropH (1).

Next, we prove that one can recover CEpropA (1) from AllSet (6). We once again choose $f_{\mathcal{V}\to\mathcal{E}}$ and $f_{\mathcal{E}\to\mathcal{V}}$ to be the sums over appropriate input multisets and ignore the second input term. This results in the following update equations:

$$\mathbf{Z}_{e,:}^{(t+1),v} = f_{\mathcal{V}\to\mathcal{E}}(V_{e\backslash v,\mathbf{X}^{(t)}}; \mathbf{Z}_{e,:}^{(t),v}) = \sum_{x\in V_{e\backslash v,\mathbf{X}^{(t)}}} x = \sum_{u\in e\backslash v} \mathbf{X}_{u,:}^{(t)}, \qquad (12)$$

$$\mathbf{X}_{v,:}^{(t+1)} = f_{\mathcal{E}\to\mathcal{V}}(E_{v,\mathbf{Z}^{(t+1),v}}; \mathbf{X}_{v,:}^{(t)}) = \sum_{x\in E_{v,\mathbf{Z}^{(t+1),v}}} x = \sum_{e:v\in e} \mathbf{Z}_{e,:}^{(t+1),v}. \qquad (13)$$

Combining the two equations we obtain

$$\mathbf{X}_{v,:}^{(t+1)} = \sum_{e:v\in e} \mathbf{Z}_{e,:}^{(t+1),v} = \sum_{e:v\in e}\sum_{u\in e\backslash v} \mathbf{X}_{u,:}^{(t)}, \qquad (14)$$

which represents the update rule of CEpropA from (1).

In the last step, we prove that one can recover Zprop (4) from AllSet (6). By choosing $f_{\mathcal{V}\to\mathcal{E}}$ and $f_{\mathcal{E}\to\mathcal{V}}$ to be the product and sum over appropriate input multisets, respectively, and by ignore the second input, we have:

$$\mathbf{Z}_{e,:}^{(t+1),v} = f_{\mathcal{V}\to\mathcal{E}}(V_{e\backslash v,\mathbf{X}^{(t)}}; \mathbf{Z}_{e,:}^{(t),v}) = \prod_{x\in V_{e\backslash v,\mathbf{X}^{(t)}}} x = \prod_{u\in e\backslash v} \mathbf{X}_{u,:}^{(t)}, \qquad (15)$$

$$\mathbf{X}_{v,:}^{(t+1)} = f_{\mathcal{E}\to\mathcal{V}}(E_{v,\mathbf{Z}^{(t+1),v}}; \mathbf{X}_{v,:}^{(t)}) = \sum_{x\in E_{v,\mathbf{Z}^{(t+1),v}}} x = \sum_{e:v\in e} \mathbf{Z}_{e,:}^{(t+1),v}. \qquad (16)$$

Combining these two equations we arrive at

$$\mathbf{X}_{v,:}^{(t+1)} = \sum_{e:v\in e} \mathbf{Z}_{e,:}^{(t+1),v} = \sum_{e:v\in e}\prod_{u\in e\backslash v} \mathbf{X}_{u,:}^{(t)}, \qquad (17)$$

which matches Zprop from (4). This completes the proof.

## D    PROOF OF THEOREM 3.4

First we prove that the hypergraph neural network layer of HGNN (2) is a special instance of AllSet (5). By choosing $f_{\mathcal{V}\to\mathcal{E}}$ and $f_{\mathcal{E}\to\mathcal{V}}$ to be the weighted sum over appropriate input multisets, where the weights are chosen according to the node degree, edge degree, and edge weights, and by ignore the second input, we have:

$$\mathbf{Z}_{e,:}^{(t+1)} = f_{\mathcal{V}\to\mathcal{E}}(V_{e,\mathbf{X}^{(t)}}; \mathbf{Z}_{e,:}^{(t)}) = \sum_{u\in e} \frac{\mathbf{X}_{u,:}^{(t)}}{\sqrt{d_u}}, \qquad (18)$$

$$\mathbf{X}_{v,:}^{(t+1)} = f_{\mathcal{E}\to\mathcal{V}}(E_{v,\mathbf{Z}^{(t+1)}}; \mathbf{X}_{v,:}^{(t)}) = \sigma\left(\frac{1}{\sqrt{d_v}} \sum_{e:v\in e} \frac{w_e \mathbf{Z}_{e,:}^{(t+1)}\mathbf{\Theta}^{(t)}}{|e|} + \mathbf{b}^{(t)}\right). \qquad (19)$$

Since $f_{\mathcal{V} \rightarrow \mathcal{E}}$ and $f_{\mathcal{E} \rightarrow \mathcal{V}}$ both use the hypergraph $\mathcal{G}$ as their input, the functions can use the node degrees, edge degrees and edge weights as their arguments. Also, it is not hard to see that our choices for $f_{\mathcal{V} \rightarrow \mathcal{E}}$ (18) and $f_{\mathcal{E} \rightarrow \mathcal{V}}$ (19) are permutation invariant in $V_{e, \mathbf{X}^{(t)}}$ and $E_{v, \mathbf{Z}^{(t+1)}}$, respectively. Plugging (18) into (19) leads to:

$$\mathbf{X}_{v,:}^{(t+1)} = \sigma \left( \frac{1}{\sqrt{d_v}} \sum_{e:v \in e} \frac{w_e}{|e|} \sum_{u \in e} \frac{\mathbf{X}_{u,:}^{(t)}}{\sqrt{d_u}} \mathbf{\Theta}^{(t)} + \mathbf{b}^{(t)} \right), \tag{20}$$

which represents the HGNN layer (2).

For the HCHA layer, we have a node-wise formulation that reads as follows:

$$\mathbf{X}_{v,:}^{(t+1)} = \sigma \left( \left[ \frac{1}{d_v} \sum_{e:v \in e} \frac{w_e \alpha_{ve}^{(t)}}{|e|} \sum_{u:u \in e} \alpha_{ue}^{(t)} \mathbf{X}_{u,:}^{(t)} \right] \mathbf{\Theta}^{(t)} + \mathbf{b}^{(t)} \right), \tag{21}$$

where the attention weight $\alpha_{ue}^{(t)}$ depends on the node features $\mathbf{X}^{(t)}$ and the feature of hyperedge $e$. Here, $\sigma(\cdot)$ is a nonlinear activation function such as LeakyReLU and eLU. An analysis similar to that described for HGNN can be used in this case as well. The only difference is in the choice of the attention weights $\alpha_{ue}$ and $\alpha_{ve}$. The definition of attention weights for HCHA is presented below.

$$\alpha_{ue}^{(t)} = \frac{\exp(\sigma(\mathbf{a}^T [\mathbf{X}_{u,:}^{(t)} \| \mathbf{Z}_{e,:}^{(t)}]))}{\sum_{e:u \in e} \exp(\sigma(\mathbf{a}^T [\mathbf{X}_{u,:}^{(t)} \| \mathbf{Z}_{e,:}^{(t)}]))}. \tag{22}$$

Clearly, the attention function in (22) has $V_{e, \mathbf{X}^{(t)}}$ and $\mathbf{Z}_{e,:}^{(t)}$ as its arguments. Furthermore, we can choose $f_{\mathcal{V} \rightarrow \mathcal{E}}$ and $f_{\mathcal{E} \rightarrow \mathcal{V}}$ as:

$$\mathbf{Z}_{e,:}^{(t+1)} = f_{\mathcal{V} \rightarrow \mathcal{E}}(V_{e, \mathbf{X}^{(t)}}; \mathbf{Z}_{e,:}^{(t)}) = \sum_{u \in e} \alpha_{ue}^{(t)} \mathbf{X}_{u,:}^{(t)}, \tag{23}$$

$$\mathbf{X}_{v,:}^{(t+1)} = f_{\mathcal{E} \rightarrow \mathcal{V}}(E_{v, \mathbf{Z}^{(t+1)}}; \mathbf{X}_{v,:}^{(t)}) = \sigma \left( \frac{1}{\sqrt{d_v}} \sum_{e:v \in e} \frac{w_e \alpha_{ve}^{(t)} \mathbf{Z}_{e,:}^{(t+1)} \mathbf{\Theta}^{(t)}}{|e|} + \mathbf{b}^{(t)} \right). \tag{24}$$

Note that both (23) and (24) are permutation invariant, which means that they are valid choices for $f_{\mathcal{V} \rightarrow \mathcal{E}}$ and $f_{\mathcal{E} \rightarrow \mathcal{V}}$. Combining these two equations reproduces the HCHA layer of (21).

For the HyperGCN layer, ignoring the degree normalization, one can use the following formulation:

$$\mathbf{X}_{v,:}^{(t+1)} = \sigma \left( \left[ \sum_{e:v \in e} \sum_{u:u \in e} w_{uv,e}^{(t)} \mathbf{X}_{u,:}^{(t)} \right] \mathbf{\Theta}^{(t)} + \mathbf{b}^{(t)} \right). \tag{25}$$

Here, the weight $w_{uv,e}^{(t)}$ depends on all node features within the hyperedge $\{\mathbf{X}_{u,:}^{(t)} : u \in e\}$, and $\sigma(\cdot)$ is a nonlinear activation function (for example, ReLU). The same analysis as presented for the previous cases may be used in this case as well: The only difference lies in the choice of the weights $w_{uv,e}^{(t)}$. According to the original HyperGCN paper (Yadati et al., 2019), these weights are defined as

$$w_{uv,e}^{(t)} = \begin{cases} \frac{1}{2|e|-3} & \text{if } u \in \{i_e, j_e\} \text{ or } v \in \{i_e, j_e\}, \\ 0 & \text{otherwise.} \end{cases}, \tag{26}$$

$$\text{where } (i_e, j_e) = \operatorname{argmax}_{u,v \in e} \|(\mathbf{X}_{u,:}^{(t)} - \mathbf{X}_{v,:}^{(t)}) \mathbf{\Theta}^{(t)}\|. \tag{27}$$

Again, it is straightforward to see that $w_{uv,e}^{(t)}$ is a function of $V_{e \setminus v, \mathbf{X}^{(t)}}$ and $\mathbf{X}_{v,:}^{(t)}$. Also, it is permutation invariant with respect to $V_{e \setminus v, \mathbf{X}^{(t)}}$. Hence, we can choose $f_{\mathcal{V} \rightarrow \mathcal{E}}$ and $f_{\mathcal{E} \rightarrow \mathcal{V}}$ as

$$\mathbf{Z}_{e,:}^{(t+1),v} = f_{\mathcal{V} \rightarrow \mathcal{E}}(V_{e \setminus v, \mathbf{X}^{(t)}}; \mathbf{Z}_{e,:}^{(t),v}, \mathbf{X}_{v,:}^{(t)}) = \sum_{u \in e \setminus v} w_{uv,e}^{(t)} \mathbf{X}_{u,:}^{(t)}, \tag{28}$$

$$\mathbf{X}_{v,:}^{(t+1)} = f_{\mathcal{E} \rightarrow \mathcal{V}}(E_{v, \mathbf{Z}^{(t+1),v}}; \mathbf{X}_{v,:}^{(t)}) = \sigma \left( \sum_{e:v \in e} \mathbf{Z}_{e,:}^{(t+1),v} \mathbf{\Theta}^{(t)} + \mathbf{b}^{(t)} \right). \tag{29}$$

Combining the two equations leads to the HyperGCN layer from (25).

Next, we show that HNHN layer introduced in Dong et al. (2020) is a special case of AllSet (5). The definition of HNHN layer is as follows

$$\mathbf{Z}_{e,:}^{(t+1)} = \sigma\left(\left[\frac{1}{d_{e,l,\beta}}\sum_{u\in e}d_{u,r,\beta}\mathbf{X}_{u,:}^{(t)}\right]\boldsymbol{\Theta}_{\mathcal{E}}^{(t)} + \mathbf{b}_{\mathcal{E}}^{(t)}\right), \tag{30}$$

$$\mathbf{X}_{v,:}^{(t+1)} = \sigma\left(\left[\frac{1}{d_{v,l,\alpha}}\sum_{e:v\in e}d_{e,r,\alpha}\mathbf{Z}_{e,:}^{(t+1)}\right]\boldsymbol{\Theta}_{\mathcal{V}}^{(t)} + \mathbf{b}_{\mathcal{V}}^{(t)}\right), \tag{31}$$

$$\text{where } d_{e,l,\beta} = \sum_{u\in e}|d_u|^\beta, \; d_{u,r,\beta} = d_u^\beta; \; d_{v,l,\beta} = \sum_{e:v\in e}|d_v|^\alpha, \; d_{v,r,\alpha} = d_v^\alpha. \tag{32}$$

Note that $\alpha$ and $\beta$ are two hyperparameters that can be tuned in HNHN. As before, $\sigma(\cdot)$ is a non-linear activation function (e.g., ReLU). It is obvious that we can choose $f_{\mathcal{V}\to\mathcal{E}}$ and $f_{\mathcal{E}\to\mathcal{V}}$ according to (30) and (31), respectively. This is due to the fact that these expressions only involves degree normalizations and linear transformations.

Finally, we show that the HyperSAGE layer from Arya et al. (2020) is a special case of AllSet (5). The HyperSAGE layer update rules are as follows:

$$\mathbf{Z}_{e,:}^{(t+1)} = \left(\frac{1}{|e|}\sum_{u\in e}(\mathbf{X}_{u,:}^{(t)})^p\right)^{1/p}, \tag{33}$$

$$\mathbf{X}_{v,:}^{(t+1),\star} = \left(\frac{1}{|\{e:v\in e\}|}\sum_{e:v\in e}(\mathbf{Z}_{e,:}^{(t)})^p\right)^{1/p} + \mathbf{X}_{v,:}^{(t)}, \tag{34}$$

$$\mathbf{X}_{v,:}^{(t+1)} = \sigma\left(\frac{\mathbf{X}_{v,:}^{(t+1),\star}}{||\mathbf{X}_{v,:}^{(t+1),\star}||}\boldsymbol{\Theta}^{(t+1)}\right). \tag{35}$$

where $\sigma(\cdot)$ is a nonlinear activation function. The update (33) can be recovered by simply choosing $f_{\mathcal{V}\to\mathcal{E}}$ to be the $l_p$ (power) mean. For the $f_{\mathcal{E}\to\mathcal{V}}$ component, we first model $f_{\mathcal{E}\to\mathcal{V}}$ as a composition of another two multiset functions. The first is the $l_p$ (power) mean with respect to its first input, while the second is addition with respect to the second input. This recovers (34). The second of the two defining functions can be chosen as (35), which is also a multiset function. This procedure leads to $f_{\mathcal{E}\to\mathcal{V}}$. This completes the proof of the first claim pertaining to the universality of AllSet.

To address the claim that the above described hypergrpah neural network layers have strictly smaller expressive power then AllSet, one only has to observe that these hypergraph neural network layers cannot approximate arbitrary multiset functions in either the $f_{\mathcal{V}\to\mathcal{E}}$ or $f_{\mathcal{E}\to\mathcal{V}}$ component. We discuss both these functions separately.

With regards to the HGNN layer, it is clear that two linear transformations cannot model arbitrary multiset functions such as the product in Zprop (4). For the HCHA layer, we note that if all node features are identical and the hyperedge features are all-zero vectors, all the attention weights $\alpha_{ue}$ are the same. This setting is similar to that described for the HGNN layer and thus this rule cannot model arbitrary multiset functions. For the HyperGCN layer (25), consider a 3-uniform hypergraph as its input. In this case, by definition, the weights $w_{uv,e}$ are all equal. Again, using only linear transformations one cannot model the product operation in Zprop (4); the same argument holds even when degree normalizations are included. For the HNHN layer, note that the $\mathcal{E}\to\mathcal{V}$ component of (31) is just one MLP layer that follows the sum $\sum_{e:v\in e}$, which cannot model arbitrary multiset functions based on the results of Deep Sets (Zaheer et al., 2017). As a final note, we point out that the HyperSAGE layer involves only one learnable matrix $\boldsymbol{\Theta}$ and is hence also a linear transformation, which cannot model arbitrary multiset functions.

This completes the proof.

## E    PROOF OF THEOREM 3.5

The propagation rule of MPNN reads as follows:

$$\mathbf{m}_{v,:}^{(t+1)} = \sum_{u \in N(v)} M_t(\mathbf{X}_{u,:}^{(t)}, \mathbf{X}_{v,:}^{(t)}, \mathbf{Z}_{e,:}^{(0),v}), \quad \mathbf{X}_{v,:}^{(t+1)} = U_t(\mathbf{X}_{v,:}^{(t)}, \mathbf{m}_{v,:}^{(t+1)}). \tag{36}$$

Here, $\mathbf{m}$ denotes the message obtained by aggregating messages from the neighborhood of the node $v$, while $M_t$ and $U_t$ are certain functions selected at the $t$-th step of propagation. To recover MPNN from AllSet (6), we simply choose $f_{\mathcal{V} \to \mathcal{E}}$ and $f_{\mathcal{E} \to \mathcal{V}}$ as

$$\mathbf{Z}_{e,:}^{(t+1),v} \triangleq f_{\mathcal{V} \to \mathcal{E}}(V_{e \setminus v, \mathbf{X}^{(t)}}; \mathbf{Z}_{e,:}^{(t),v}) = f_{\mathcal{V} \to \mathcal{E}}(\{\mathbf{X}_{u,:}^{(t)}\}; \mathbf{Z}_{e,:}^{(t),v}) = \mathbf{X}_{u,:}^{(t)} \parallel \mathbf{Z}_{e,:}^{(0),v}, \quad \text{s.t. } u \in e \setminus v,$$

$$\mathbf{X}_{v,:}^{(t+1)} \triangleq f_{\mathcal{E} \to \mathcal{V}}(E_{v, \mathbf{Z}^{(t+1),v}}; \mathbf{X}_{v,:}^{(t)}) = f_{\mathcal{E} \to \mathcal{V}}(\{\mathbf{Z}_{e,:}^{(t+1),v}\}_{e:v \in e}; \mathbf{X}_{v,:}^{(t)})$$

$$= U_t \left( \mathbf{X}_{v,:}^{(t)}, \sum_{e:v \in e} M_t'(\mathbf{Z}_{e,:}^{(t+1),v}, \mathbf{X}_{v,:}^{(t)}) \right) = U_t \left( \mathbf{X}_{v,:}^{(t)}, \sum_{e:v \in e} M_t(\mathbf{X}_{u,:}^{(t)}, \mathbf{X}_{v,:}^{(t)}, \mathbf{Z}_{e,:}^{(0),v}) \right). \tag{37}$$

Hence, MPNN is a special case of AllSet for graph inputs.

## F    PROOF OF PROPOSITION 4.1

The proof is largely based on the proof of Proposition 2 of the Set Trasnformer paper (Lee et al., 2019) but is nevertheless included for completeness. We ignore all layer normalizations as in the proof of (Lee et al., 2019) and start with the following theorem.

**Theorem F.1** (Theorem 1 in Lee et al. (2019)). *Functions of the form MLP $\sum$ (MLP) are universal approximators in the space of permutation invariant functions.*

This result is based on Zaheer et al. (2017). According to Wagstaff et al. (2019), we have the constraint that the input multiset has to be finite.

Next, we show that the simple mean operator is a special case of a multihead attention module. For simplicity, we consider the case of a single head, which corresponds to $h = 1$, as we can always set the other heads to zero by choosing $\text{MLP}^{V,i}(\mathbf{S}) = 0$ for all $i \geq 2$. Following the proof of Lemma 3 in Lee et al. (2019), we set the learnable weight $\theta$ to be the all-zero vector and use $\omega(\cdot) = 1 + g(\cdot)$ (element-wise) as an activation function such that $g(0) = 0$. The $\text{MH}_{1,\omega}$ module in (8) then becomes

$$\text{MH}_{1,\omega}(\theta, \mathbf{S}, \mathbf{S}) = \mathbf{O}^{(1)} = \omega \left( \theta(\text{MLP}^{K,1}(\mathbf{S}))^T \right) \text{MLP}^{V,1}(\mathbf{S}) = \sum_{i=1}^{|S|} \text{MLP}^{V,1}(\mathbf{S})_i. \tag{38}$$

The AllSetTranformer layer (8) takes the form

$$f_{\mathcal{V} \to \mathcal{E}}(S) = \mathbf{Y} + \text{MLP}(\mathbf{Y}), \text{ where } \mathbf{Y} = \sum_{i=1}^{|S|} \text{MLP}^{V,1}(\mathbf{S}). \tag{39}$$

Note that $\mathbf{Y}$ is a vector. Clearly, we can choose an MLP such that $\text{MLP}(\mathbf{Y})$ equals to another $\text{MLP}(\mathbf{Y})$ that results in subtracting $\mathbf{Y}$. Thus, we have:

$$f_{\mathcal{V} \to \mathcal{E}}(S) = \text{MLP}(\mathbf{Y}) = \text{MLP}\left( \sum_{i=1}^{|S|} \text{MLP}^{V,1}(\mathbf{S}) \right). \tag{40}$$

By Theorem F.1, it is clear that $f_{\mathcal{V} \to \mathcal{E}}(S)$ is a universal approximator for permutation invariant functions. The same analysis applies to $f_{\mathcal{E} \to \mathcal{V}}(S)$.

## G    A DISCUSSION OF LEGCN

Line expansion (LE) is a procedure that transform a hypergraph or a graph into a homogeneous graph; a node in the LE graph represents a pair of node-hyperedge from the original hypergraph.

Nodes in the LE graph are linked if and only if there is a nonempty intersection of their associated node-hyperedge pairs. As stated by the authors of LEGCN (Yang et al., 2020), for a fully connected $d$-uniform hypergraph their resulting LE graph has $\Theta(d|\mathcal{E}|)$ nodes and $\Theta(d^2|\mathcal{E}|^2)$ edges. This expansion hence leads to very large graphs which require large memory units and have high computational complexity when coupled with GNNs such as GCN. To alleviate this drawback, the authors use random sampling techniques which unfortunately lead to an undesired information loss. In contrast, we define our AllSet layers directly on hypergraph which leads to a significantly more efficient approach.

## H Hypergraph Propagation Rules Based on the H Eigenproblem

As outlined in the main text, in this case we associate a $d$-uniform hypergraph $\mathcal{G}$ with an adjacency tensor $\mathbf{A}$. The H eigenproblem for $\mathbf{A}$ states that

$$\mathbf{A}\mathbf{x}^{d-1} = \lambda\mathbf{x}^{[d-1]}, \quad \text{where } \mathbf{x}_i^{[d-1]} = (\mathbf{x}_i)^{d-1}. \tag{41}$$

Similar to Zprop, we can define Hprop (41) in a node-wise fashion as

$$\text{Hprop: } \mathbf{X}_{v,:}^{(t+1)} = \left( \sum_{e:v\in e} (d-1) \prod_{u:u\in e\backslash v} \mathbf{X}_{u,:}^{(t)} \right)^{\frac{1}{d-1}}. \tag{42}$$

Although taking the $1/(d-1)$-th root resolves the unit issue that exists in Zprop (4), it may lead to imaginary features during updates. It remains an open question to formulate Hprop in a manner that ensures that features remain real-valued during propagation.

## I Additional Details Pertaining to the Tested Datasets

Table 3: Full dataset statistics: $|e|$ refers to the size of the hyperedges while $d_v$ refers to the node degree.

|  | Cora | Citeseer | Pubmed | Cora-CA | DBLP-CA | Zoo | 20News | Mushroom | NTU2012 | ModelNet40 | Yelp | House | Walmart |
|---|---|---|---|---|---|---|---|---|---|---|---|---|---|
| $|\mathcal{V}|$ | 2708 | 3312 | 19717 | 2708 | 41302 | 101 | 16242 | 8124 | 2012 | 12311 | 50758 | 1290 | 88860 |
| $|\mathcal{E}|$ | 1579 | 1079 | 7963 | 1072 | 22363 | 43 | 100 | 298 | 2012 | 12311 | 679302 | 341 | 69906 |
| # feature | 1433 | 3703 | 500 | 1433 | 1425 | 16 | 100 | 22 | 100 | 100 | 1862 | 100 | 100 |
| # class | 7 | 6 | 3 | 7 | 6 | 7 | 4 | 2 | 67 | 40 | 9 | 2 | 11 |
| max $|e|$ | 5 | 26 | 171 | 43 | 202 | 93 | 2241 | 1808 | 5 | 5 | 2838 | 81 | 25 |
| min $|e|$ | 2 | 2 | 2 | 2 | 2 | 1 | 29 | 1 | 5 | 5 | 2 | 1 | 2 |
| avg $|e|$ | 3.03 | 3.2 | 4.35 | 4.28 | 4.45 | 39.93 | 654.51 | 136.31 | 5 | 5 | 6.66 | 34.72 | 6.59 |
| med $|e|$ | 3 | 2 | 3 | 3 | 3 | 40 | 537 | 72 | 5 | 5 | 3 | 40 | 5 |
| max $d_v$ | 145 | 88 | 99 | 23 | 18 | 17 | 44 | 5 | 19 | 30 | 7855 | 44 | 5733 |
| min $d_v$ | 0 | 0 | 0 | 0 | 1 | 17 | 1 | 5 | 1 | 1 | 1 | 0 | 0 |
| avg $d_v$ | 1.77 | 1.04 | 1.76 | 1.69 | 2.41 | 17 | 4.03 | 5 | 5 | 5 | 89.12 | 9.18 | 5.18 |
| med $d_v$ | 1 | 0 | 0 | 2 | 2 | 17 | 3 | 5 | 5 | 4 | 35 | 7 | 2 |

We use 10 available benchmark datasets from the existing hypergraph neural networks literature and introduce three newly created datasets as described in the exposition to follow. The benchmark datasets include cocitation networks Cora, Citeseer and Pubmed[5], obtained from Yadati et al. (2019). The coauthorship networks Cora-CA[6] and DBLP-CA[7] are also adapted from Yadati et al. (2019). In the cocitation and coauthorship networks datasets, the node features are the bag-of-words representations of the corresponding documents. Datasets from the UCI Categorical Machine Learning Repository (Dua & Graff, 2017) include 20Newsgroups, Mushroom and Zoo. In 20Newsgroups, the node features are the TF-IDF representations of news messages. In the Mushroom dataset, the node features represent categorical descriptions of 23 species of mushrooms. In Zoo, the node features are a mix of categorical and numerical measurements describing different animals. The computer vision and graphics datasets include the Princeton CAD ModelNet40 (Wu et al., 2015) and the NTU2012 3D dataset (Chen et al., 2003). The visual objects contain features extracted using Group-View Convolutional Neural Network(GVCNN) (Feng et al., 2018) and Multi-View Convolutional Neural

---

[5] https://linqs.soe.ucsc.edu/data     [6] https://people.cs.umass.edu/     mccallum/data.html
[7] https://aminer.org/lab-datasets/citation/DBLP-citation-Jan8.tar.bz

Network(MVCNN) (Su et al., 2015). The hypergraph construction follows the setting described in Feng et al. (2019); Yang et al. (2020).

The three newly introduce datasets are adapted from Yelp (Yelp), House (Chodrow et al., 2021) and Walmart (Amburg et al., 2020). For Yelp, we selected all businesses in the "restaurant" catalog as our nodes, and formed hyperedges by selecting restaurants visited by the same user. We use the number of stars in the average review of a restaurant as the corresponding node label, starting from $1$ and going up to $5$ stars, with an interval of $0.5$ stars. We then form the node features from the latitude, longitude, one-hot encoding of city and state, and bag-of-word encoding of the top-1000 words in the name of the corresponding restaurants. In the House dataset, each node is a member of the US House of Representatives and hyperedges are formed by grouping together members of the same committee. Node labels indicate the political party of the representatives. In Walmart, nodes represent products being purchased at Walmart, and hyperedges equal sets of products purchased together; the node labels are the product categories. Since there are no node features in the original House and Walmart dataset, we impute the same using Gaussian random vectors, in a manner similar to what was done for the contextual stochastic block model (Deshpande et al., 2018). In both datasets, we fix the feature vector dimension to $100$ and use one-hot encodings of the labels with added Gaussian noise $\mathcal{N}(0, \sigma^2 \mathbf{I})$ as the actual features. The noise standard deviation $\sigma$ is chosen to be $1$ and $0.6$.

## J COMPUTATIONAL EFFICIENCY AND EXPERIMENTAL SETTINGS

All our test were executed on a Linux machine with $48$ cores, 376GB of system memory, and two NVIDIA Tesla P100 GPUs with 12GB of GPU memory each. In AllSetTransformer, we also used a single layer MLP at the end for node classification. The average training times with their corresponding standard deviations in second per run, for all baseline methods at all tested datasets, are reported in Table 4. Note that the reported times do not include any preprocessing times for the hypergraph datasets as these are only performed once before training. Table 4 lists the average training time per run with the optimal set of hyperparameters obtained after tunning; the average training times per run for different sets of hyperparameters used in the tuning process are reported in Table 5.

**Choices of hyperparameters.** We tune the hidden dimension of all hypergraph neural networks over $\{64, 128, 256, 512\}$, except for the case of HyperGCN and HAN, where the original implementations do not allow for changing the hidden dimension. For HNHN, AllSetTransformer and AllDeepSets we use one layer (a full $\mathcal{V} \rightarrow \mathcal{E} \rightarrow \mathcal{V}$ propagation rule layer), while for all the other methods we use two layers as recommended in the previous literature. We tune the learning rate over $\{0.1, 0.01, 0.001\}$, and the weight decays over $\{0, 0.00001\}$, for all models under consideration. For models with multihead attentions, we also tune the number of heads over the set $\{1, 4, 8\}$. The best hyperparameters for each model and dataset are listed in Table 6. Note that we only use default setting for HAN due to facts that its DGL implementation has all parameters set as constants and its much higher time complexity per run.

Table 4: Running times for the best hyperparameter choice: Mean (sec or hour) $\pm$ standard deviation.

| | Cora | Citeseer | Pubmed | Cora-CA | DBLP-CA | Zoo |
|---|---|---|---|---|---|---|
| AllSetTransformer | 7.37s $\pm$ 0.37s | 15.97s $\pm$ 0.17s | 26.82s $\pm$ 0.13s | 6.37s $\pm$ 0.13s | 134.96s $\pm$ 0.37s | 5.36s $\pm$ 0.13s |
| AllDeepSets | 11.70s $\pm$ 0.07s | 14.56s $\pm$ 0.06s | 58.26s $\pm$ 0.23s | 11.23s $\pm$ 0.06s | 145.60s $\pm$ 11.00s | 9.57s $\pm$ 0.75s |
| MLP | 1.33s $\pm$ 0.06s | 1.52s $\pm$ 0.04s | 1.77s $\pm$ 0.05s | 1.38s $\pm$ 0.06s | 3.86s $\pm$ 0.02s | 0.03s $\pm$ 0.00s |
| CECGN | 2.39s $\pm$ 0.09s | 2.06s $\pm$ 0.07s | 3.42s $\pm$ 0.30s | 1.68s $\pm$ 0.05s | 9.19s $\pm$ 0.04s | 1.69s $\pm$ 0.10s |
| CEGAT | 10.84s $\pm$ 0.21s | 5.34s $\pm$ 0.12s | 23.20s $\pm$ 0.15s | 4.86s $\pm$ 0.13s | 69.50s $\pm$ 0.38s | 3.92s $\pm$ 0.13s |
| HNHN | 2.71s $\pm$ 0.04s | 2.84s $\pm$ 0.03s | 8.92s $\pm$ 0.03s | 2.67s $\pm$ 0.05s | 26.57s $\pm$ 0.07s | 0.19s $\pm$ 0.02s |
| HGNN | 4.81s $\pm$ 0.17s | 5.03s $\pm$ 0.16s | 14.75s $\pm$ 0.08s | 3.32s $\pm$ 0.15s | 21.65s $\pm$ 0.09s | 3.18s $\pm$ 0.04s |
| HCHA | 3.57s $\pm$ 0.16s | 4.02s $\pm$ 0.05s | 14.26s $\pm$ 0.06s | 3.18s $\pm$ 0.29s | 37.49s $\pm$ 0.08s | 2.87s $\pm$ 0.09s |
| HyperGCN | 2.91s $\pm$ 0.02s | 3.37s $\pm$ 0.02s | 4.33s $\pm$ 0.27s | 2.97s $\pm$ 0.03s | 8.07s $\pm$ 0.21s | N/A |
| UniGCNII | 43.21s $\pm$ 0.06s | 2.78s $\pm$ 0.27s | 8.93s $\pm$ 0.83s | 16.55s $\pm$ 0.05s | 223.57s $\pm$ 0.03s | 2.35s $\pm$ 0.12s |
| HAN (full batch) | 3.47s $\pm$ 0.52s | 2.58s $\pm$ 0.48s | 7.17s $\pm$ 0.11s | 3.09s $\pm$ 0.07s | 12.39s $\pm$ 0.22s | 2.78s $\pm$ 0.09s |
| HAN (mini batch) | 99.87s $\pm$ 17.05s | 69.06s $\pm$ 11.69s | 502.44s $\pm$ 161.40s | 143.80s $\pm$ 15.48s | 0.45h $\pm$ 0.12h | 102.98s $\pm$ 2.39s |

| | 20Newsgroups | Mushroom | NTU2012 | ModelNet40 | Yelp | House(1) | Walmart(1) |
|---|---|---|---|---|---|---|---|
| AllSetTransformer | 25.47s $\pm$ 0.50s | 10.12s $\pm$ 0.14s | 7.40s $\pm$ 0.28s | 47.55s $\pm$ 0.25s | 284.79s $\pm$ 0.23s | 8.69s $\pm$ 0.32s | 154.19s $\pm$ 0.11s |
| AllDeepSets | 46.61s $\pm$ 0.14s | 14.07s $\pm$ 0.03s | 10.25s $\pm$ 0.07s | 49.27s $\pm$ 0.13s | 474.34s $\pm$ 0.07s | 4.07s $\pm$ 0.24s | 341.18s $\pm$ 0.62s |
| MLP | 1.68s $\pm$ 0.02s | 1.44s $\pm$ 0.06s | 1.29s $\pm$ 0.01s | 1.60s $\pm$ 0.13s | 5.70s $\pm$ 0.03s | 1.51s $\pm$ 0.05s | 7.83s $\pm$ 0.02s |
| CECGN | OOM | 69.67s $\pm$ 1.70s | 2.00s $\pm$ 0.02s | 2.54s $\pm$ 0.28s | OOM | 10.40s $\pm$ 0.08s | 173.75s $\pm$ 0.07s |
| CEGAT | OOM | 132.53s $\pm$ 27.81s | 8.44s $\pm$ 0.12s | 43.57s $\pm$ 16.13s | OOM | 26.38s $\pm$ 0.15s | 121.90s $\pm$ 0.11s |
| HNHN | 9.04s $\pm$ 0.09s | 2.34s $\pm$ 0.03s | 2.17s $\pm$ 0.03s | 8.54s $\pm$ 0.08s | 111.00s $\pm$ 0.08s | 1.76s $\pm$ 0.01s | 54.85s $\pm$ 0.05s |
| HGNN | 0.51s $\pm$ 0.03s | 10.57s $\pm$ 0.10s | 3.46s $\pm$ 0.09s | 15.55s $\pm$ 0.13s | 550.93s $\pm$ 0.07s | 3.20s $\pm$ 0.13s | 105.68s $\pm$ 0.05s |
| HCHA | 0.53s $\pm$ 0.03s | 10.54s $\pm$ 0.13s | 3.47s $\pm$ 0.11s | 15.95s $\pm$ 0.81s | 267.83s $\pm$ 0.03s | 3.05s $\pm$ 0.08s | 104.27s $\pm$ 0.14s |
| HyperGCN | 5.96s $\pm$ 0.03s | 5.01s $\pm$ 0.03s | 3.09s $\pm$ 0.02s | 4.77s $\pm$ 0.02s | 183.76s $\pm$ 0.82s | 2.98s $\pm$ 0.04s | 16.03s $\pm$ 0.34s |
| UniGCNII | 34.58s $\pm$ 0.03s | 10.00s $\pm$ 0.03s | 12.95s $\pm$ 0.23s | 4.40s $\pm$ 0.16s | 197.07s $\pm$ 0.04s | 12.10s $\pm$ 0.06s | 104.23s $\pm$ 0.10s |
| HAN (full batch) | OOM | 30.28s $\pm$ 3.29s | 3.10s $\pm$ 0.11s | 4.56s $\pm$ 0.21s | OOM | 3.32s $\pm$ 0.12s | OOM |
| HAN (mini batch) | 313.28s $\pm$ 93.42s | 111.53s $\pm$ 30.05s | 165.35s $\pm$ 1.96s | 414.50s $\pm$ 87.25s | 5.50h $\pm$ 1.48h | 72.79s $\pm$ 5.59s | 1.96h $\pm$ 0.44h |

Table 5: Average running times over all different choices of hyperparameters: Mean $\pm$ standard deviation.

| | Cora | Citeseer | Pubmed | Cora-CA | DBLP-CA | Zoo |
|---|---|---|---|---|---|---|
| AllSetTransformer | 8.18s $\pm$ 3.59s | 8.41s $\pm$ 4.39s | 22.99s $\pm$ 16.54s | 7.21s $\pm$ 2.94s | 53.14s $\pm$ 43.49s | 5.97s $\pm$ 2.23s |
| AllDeepSets | 5.61s $\pm$ 4.47s | 6.83s $\pm$ 5.63s | 27.05s $\pm$ 23.94s | 5.62s $\pm$ 4.38s | 57.26s $\pm$ 55.65s | 4.14s $\pm$ 3.18s |
| MLP | 1.00s $\pm$ 0.46s | 1.03s $\pm$ 0.48s | 1.53s $\pm$ 0.95s | 1.00s $\pm$ 0.45s | 1.53s $\pm$ 1.25s | 0.99s $\pm$ 0.50s |
| CEGCN | 1.63s $\pm$ 0.76s | 2.24s $\pm$ 1.38s | 5.93s $\pm$ 5.01s | 1.79s $\pm$ 0.99s | 18.91s $\pm$ 17.08s | 1.41s $\pm$ 0.62s |
| CEGAT | 6.47s $\pm$ 3.99s | 9.25s $\pm$ 7.60s | 34.52s $\pm$ 35.16s | 7.93s $\pm$ 5.94s | 53.84s $\pm$ 37.83s | 5.28s $\pm$ 5.92s |
| HNHN | 1.65s $\pm$ 1.01s | 2.43s $\pm$ 1.62s | 5.03s $\pm$ 3.14s | 1.89s $\pm$ 0.80s | 14.80s $\pm$ 9.55s | 1.59s $\pm$ 0.63s |
| HGNN | 3.42s $\pm$ 1.49s | 4.44s $\pm$ 2.03s | 8.44s $\pm$ 5.08s | 3.37s $\pm$ 1.47s | 21.02s $\pm$ 13.61s | 2.75s $\pm$ 1.06s |
| HCHA | 3.32s $\pm$ 1.42s | 4.30s $\pm$ 2.02s | 8.22s $\pm$ 4.97s | 3.24s $\pm$ 1.42s | 20.66s $\pm$ 13.37s | 2.65s $\pm$ 1.03s |
| HyperGCN | 2.13s $\pm$ 0.75s | 2.48s $\pm$ 0.76s | 3.23s $\pm$ 1.00s | 2.27s $\pm$ 0.67s | 6.13s $\pm$ 1.93s | 1.91s $\pm$ 0.58s |
| UniGCNII | 13.13s $\pm$ 13.64s | 16.81s $\pm$ 17.63s | 61.81s $\pm$ 75.63s | 11.61s $\pm$ 12.46s | 94.97s $\pm$ 81.68s | 3.56s $\pm$ 2.58s |
| HAN (full batch) | 3.47s $\pm$ 0.52s | 2.58s $\pm$ 0.48s | 7.17s $\pm$ 0.11s | 3.09s $\pm$ 0.07s | 12.39s $\pm$ 0.22s | 2.78s $\pm$ 0.09s |
| HAN (mini batch) | 99.87s $\pm$ 17.05s | 69.06s $\pm$ 11.69s | 502.44s $\pm$ 161.40s | 143.80s $\pm$ 15.48s | 0.45h $\pm$ 0.12h | 102.98s $\pm$ 2.39s |

| | 20Newsgroups | Mushroom | NTU2012 | ModelNet40 | Yelp | House(1) | Walmart(1) |
|---|---|---|---|---|---|---|---|
| AllSetTransformer | 21.97s $\pm$ 17.96s | 14.35s $\pm$ 11.10s | 6.96s $\pm$ 2.65s | 19.99s $\pm$ 14.46s | 244.61s $\pm$ 97.44s | 6.87s $\pm$ 3.11s | 123.51s $\pm$ 102.91s |
| AllDeepSets | 20.88s $\pm$ 21.73s | 12.76s $\pm$ 13.89s | 5.15s $\pm$ 4.03s | 19.90s $\pm$ 19.15s | 196.53s $\pm$ 254.83s | 4.52s $\pm$ 3.01s | 107.54s $\pm$ 119.02s |
| MLP | 1.22s $\pm$ 0.54s | 1.11s $\pm$ 0.53s | 0.97s $\pm$ 0.43s | 1.14s $\pm$ 0.50s | 3.88s $\pm$ 1.93s | 0.98s $\pm$ 0.47s | 4.44s $\pm$ 2.72s |
| CEGCN | OOM | 67.66s $\pm$ 47.17s | 1.51s $\pm$ 0.69s | 3.68s $\pm$ 2.72s | OOM | 4.16s $\pm$ 3.65s | 63.42s $\pm$ 62.30s |
| CEGAT | OOM | 121.39s $\pm$ 7.43s | 5.45s $\pm$ 2.97s | 19.54s $\pm$ 19.68s | OOM | 21.94s $\pm$ 23.44s | 85.82s $\pm$ 45.10s |
| HNHN | 5.21s $\pm$ 3.18s | 3.28s $\pm$ 1.80s | 1.68s $\pm$ 0.68s | 4.92s $\pm$ 2.99s | 56.15s $\pm$ 77.57s | 1.70s $\pm$ 0.53s | 23.62s $\pm$ 18.74s |
| HGNN | 9.69s $\pm$ 5.93s | 6.51s $\pm$ 3.58s | 3.30s $\pm$ 1.40s | 8.86s $\pm$ 5.36s | 343.16s $\pm$ 262.93s | 3.29s $\pm$ 1.21s | 44.19s $\pm$ 36.85s |
| HCHA | 9.61s $\pm$ 5.98s | 6.48s $\pm$ 3.57s | 3.23s $\pm$ 1.37s | 8.82s $\pm$ 5.47s | 135.47s $\pm$ 187.18s | 3.20s $\pm$ 1.18s | 43.62s $\pm$ 36.37s |
| HyperGCN | 4.34s $\pm$ 1.54s | 3.56s $\pm$ 1.25s | 2.29s $\pm$ 0.72s | 3.41s $\pm$ 1.17s | 125.67s $\pm$ 56.93s | 2.39s $\pm$ 0.76s | 20.11s $\pm$ 7.88s |
| UniGCNII | 58.01s $\pm$ 66.97s | 35.47s $\pm$ 44.72s | 9.22s $\pm$ 9.68s | 46.30s $\pm$ 54.77s | 309.35s $\pm$ 129.68s | 8.83s $\pm$ 7.97s | 119.66s $\pm$ 73.89s |
| HAN (full batch) | OOM | 30.28s $\pm$ 3.29s | 3.10s $\pm$ 0.11s | 4.56s $\pm$ 0.21s | OOM | 3.32s $\pm$ 0.12s | OOM |
| HAN (mini batch) | 313.28s $\pm$ 93.42s | 111.53s $\pm$ 30.05s | 165.35s $\pm$ 1.96s | 414.50s $\pm$ 87.25s | 5.50h $\pm$ 1.48h | 72.79s $\pm$ 5.59s | 1.96h $\pm$ 0.44h |

Table 6: Choice of hyperparameters for each method: lr refers to the learning rate, wd refers to the weight decaying factor, h1 refers to the dimension of MLP hidden layer and heads refers to the number of attention heads. Remarkably, not all hyperparameters are used for some models. For example, MLP, CEGCN, HNHN, HGNN, HCHA and HyperGCN does not take heads as input.

| | AllSetTransformer | | | | AllDeepSets | | | | MLP | | | | CEGCN | | | |
|---|---|---|---|---|---|---|---|---|---|---|---|---|---|---|---|---|
| | lr | wd | h1 | heads | lr | wd | h1 | heads | lr | wd | h1 | heads | lr | wd | h1 | heads |
| Cora | 0.001 | 0 | 256 | 4 | 0.001 | 0 | 512 | 1 | 0.01 | 0 | 64 | 1 | 0.001 | 0 | 512 | 1 |
| Citeseer | 0.001 | 0 | 512 | 8 | 0.001 | 0 | 512 | 1 | 0.01 | 0 | 64 | 1 | 0.001 | 0 | 128 | 1 |
| Pubmed | 0.001 | 0 | 256 | 8 | 0.001 | 0 | 512 | 1 | 0.01 | 1.00E-05 | 64 | 1 | 0.01 | 1.00E-05 | 64 | 1 |
| Cora-CA | 0.001 | 0 | 128 | 8 | 0.001 | 0 | 512 | 1 | 0.01 | 1.00E-05 | 64 | 1 | 0.01 | 0 | 64 | 1 |
| DBLP-CA | 0.001 | 0 | 512 | 8 | 0.001 | 0 | 512 | 1 | 0.01 | 0 | 64 | 1 | 0.01 | 1.00E-05 | 64 | 1 |
| Zoo | 0.01 | 1.00E-05 | 64 | 1 | 0.001 | 0 | 512 | 1 | 0.1 | 0 | 64 | 1 | 0.001 | 0 | 512 | 1 |
| 20News | 0.001 | 0 | 256 | 8 | 0.001 | 0 | 512 | 1 | 0.01 | 1.00E-05 | 64 | 1 | OOM | OOM | OOM | OOM |
| Mushroom | 0.001 | 0 | 128 | 1 | 0.001 | 0 | 256 | 1 | 0.01 | 0 | 64 | 1 | 0.01 | 1.00E-05 | 64 | 1 |
| NTU2012 | 0.001 | 0 | 256 | 1 | 0.001 | 0 | 512 | 1 | 0.01 | 1.00E-05 | 64 | 1 | 0.001 | 0 | 512 | 1 |
| ModelNet40 | 0.001 | 0 | 512 | 8 | 0.001 | 0 | 512 | 1 | 0.01 | 1.00E-05 | 64 | 1 | 0.01 | 1.00E-05 | 64 | 1 |
| Yelp | 0.001 | 0 | 64 | 1 | 0.001 | 0 | 128 | 1 | 0.01 | 1.00E-05 | 64 | 1 | OOM | OOM | OOM | OOM |
| House(1) | 0.001 | 0 | 512 | 8 | 0.01 | 1.00E-05 | 64 | 1 | 0.01 | 0 | 64 | 1 | 0.001 | 0 | 512 | 1 |
| Walmart(1) | 0.001 | 0 | 256 | 8 | 0.001 | 0 | 512 | 1 | 0.001 | 0 | 64 | 1 | 0.001 | 0 | 512 | 1 |
| House(0.6) | 0.001 | 0 | 512 | 1 | 0.001 | 0 | 128 | 1 | 0.01 | 1.00E-05 | 64 | 1 | 0.001 | 0 | 512 | 1 |
| Walmart(0.6) | 0.001 | 0 | 256 | 8 | 0.001 | 0 | 512 | 1 | 0.01 | 0 | 64 | 1 | 0.001 | 0 | 512 | 1 |

| | CEGAT | | | | HNHN | | | | HGNN | | | | HCHA | | | |
|---|---|---|---|---|---|---|---|---|---|---|---|---|---|---|---|---|
| | lr | wd | h1 | heads | lr | wd | h1 | heads | lr | wd | h1 | heads | lr | wd | h1 | heads |
| Cora | 0.001 | 0 | 256 | 8 | 0.001 | 0 | 512 | 1 | 0.001 | 0 | 512 | 1 | 0.001 | 0 | 256 | 1 |
| Citeseer | 0.001 | 0 | 64 | 4 | 0.001 | 0 | 256 | 1 | 0.001 | 0 | 256 | 1 | 0.001 | 0 | 128 | 1 |
| Pubmed | 0.01 | 1.00E-05 | 64 | 8 | 0.001 | 0 | 512 | 1 | 0.001 | 0 | 512 | 1 | 0.001 | 0 | 512 | 1 |
| Cora-CA | 0.01 | 0 | 64 | 4 | 0.001 | 0 | 512 | 1 | 0.001 | 0 | 128 | 1 | 0.001 | 0 | 128 | 1 |
| DBLP-CA | 0.01 | 1.00E-05 | 64 | 8 | 0.001 | 0 | 512 | 1 | 0.001 | 0 | 256 | 1 | 0.001 | 0 | 512 | 1 |
| Zoo | 0.001 | 1.00E-05 | 64 | 8 | 0.1 | 0 | 64 | 1 | 0.001 | 0 | 512 | 1 | 0.001 | 0 | 512 | 1 |
| 20News | OOM | OOM | OOM | OOM | 0.001 | 0 | 512 | 1 | 0.1 | 0 | 64 | 1 | 0.1 | 0 | 64 | 1 |
| Mushroom | 0.01 | 1.00E-05 | 64 | 1 | 0.001 | 0 | 128 | 1 | 0.001 | 0 | 512 | 1 | 0.001 | 0 | 512 | 1 |
| NTU2012 | 0.001 | 0 | 512 | 4 | 0.001 | 0 | 512 | 1 | 0.001 | 0 | 256 | 1 | 0.001 | 0 | 256 | 1 |
| ModelNet40 | 0.01 | 1.00E-05 | 64 | 8 | 0.001 | 0 | 512 | 1 | 0.001 | 0 | 512 | 1 | 0.001 | 0 | 512 | 1 |
| Yelp | OOM | OOM | OOM | OOM | 0.001 | 0 | 128 | 1 | 0.001 | 0 | 256 | 1 | 0.001 | 0 | 128 | 1 |
| House(1) | 0.001 | 0 | 128 | 8 | 0.01 | 1.00E-05 | 64 | 1 | 0.01 | 0 | 64 | 1 | 0.01 | 1.00E-05 | 64 | 1 |
| Walmart(1) | 0.001 | 0 | 256 | 1 | 0.001 | 0 | 512 | 1 | 0.001 | 0 | 512 | 1 | 0.001 | 0 | 512 | 1 |
| House(0.6) | 0.01 | 1.00E-05 | 64 | 8 | 0.001 | 0 | 256 | 1 | 0.001 | 0 | 512 | 1 | 0.001 | 0 | 512 | 1 |
| Walmart(0.6) | 0.001 | 0 | 256 | 1 | 0.001 | 0 | 512 | 1 | 0.001 | 0 | 512 | 1 | 0.001 | 0 | 256 | 1 |

| | HyperGCN | | | | UniGCNII | | | |
|---|---|---|---|---|---|---|---|---|
| | lr | wd | h1 | heads | lr | wd | h1 | heads |
| Cora | 0.001 | 0 | 64 | 1 | 0.001 | 0 | 512 | 8 |
| Citeseer | 0.01 | 1.00E-05 | 64 | 1 | 0.001 | 0 | 128 | 1 |
| Pubmed | 0.01 | 1.00E-05 | 64 | 1 | 0.001 | 0 | 128 | 1 |
| Cora-CA | 0.01 | 1.00E-05 | 64 | 1 | 0.001 | 0 | 512 | 4 |
| DBLP-CA | 0.01 | 1.00E-05 | 64 | 1 | 0.001 | 0 | 256 | 8 |
| Zoo | 0.001 | 0 | 128 | 1 | 0.001 | 0 | 128 | 8 |
| 20News | 0.01 | 1.00E-05 | 64 | 1 | 0.001 | 0 | 128 | 8 |
| Mushroom | 0.001 | 0 | 64 | 1 | 0.001 | 0 | 128 | 4 |
| NTU2012 | 0.01 | 1.00E-05 | 64 | 1 | 0.001 | 0 | 256 | 8 |
| ModelNet40 | 0.01 | 1.00E-05 | 64 | 1 | 0.001 | 0 | 128 | 1 |
| Yelp | 0.01 | 1.00E-05 | 64 | 1 | 0.001 | 0 | 128 | 1 |
| House(1) | 0.01 | 1.00E-05 | 64 | 1 | 0.001 | 0 | 256 | 4 |
| Walmart(1) | 0.001 | 0 | 128 | 1 | 0.001 | 0 | 512 | 1 |
| House(0.6) | 0.01 | 1.00E-05 | 64 | 1 | 0.001 | 0 | 512 | 1 |
| Walmart(0.6) | 0.01 | 1.00E-05 | 64 | 1 | 0.001 | 0 | 256 | 4 |

