# OpenReview forum: "You are AllSet: A Multiset Function Framework for Hypergraph Neural Networks"
_ICLR.cc/2022/Conference — ICLR 2022 Poster_

### Official Review · Reviewer_LMVx · 2021-10-23

**Correctness:** 3
**Technical Novelty And Significance:** 2
**Empirical Novelty And Significance:** 2
**Recommendation:** 6
**Confidence:** 5

**Main Review:**

### **Strengths**

1. **Quality of Paper**

The paper is technically sound.
The methods used (AllSetTransformer, AllDeepSets) are appropriate for the downstream task (node classification in hypergraphs).
It is interesting to see that AllSet generalises several neural networks on hypergraphs and message passing neural networks on graphs.

2. **Clarity of Presentation**

The paper is clearly written and well organised.
The appendix is comprehensive with both theoretical proofs and empirical details (e.g., dataset construction, hyperparameters, etc.).
The authors have also released the source code as part of the supplementary material.



### **Weaknesses**

1. **Originality of Contributions**

The key idea of AllSet is to use rules of the form $f_{\mathcal{V}\rightarrow\mathcal{E}}$ and $f_{\mathcal{E}\rightarrow\mathcal{V}}$ as given in equation 5.
These update rules are well-known ideas in both (i) hypergraph neural network literature [e.g., Be More with Less: Hypergraph Attention Networks for Inductive Text Classification, EMNLP'20] and (ii) bipartite graph neural network literature [e.g., Hierarchical Representation Learning for Bipartite Graphs, IJCAI'19].
Moreover, AllSetTransformer and AllDeepSets are straightforward applications of SetTransformer and DeepSets in the AllSet framework.


2. **Significance of Empirical Evaluation**

From Table 2 it is clear that the proposed methods show marginal or no improvements on all but three datasets viz., zoo, a tiny dataset, and Yelp, Walmart both curated by authors.  The Yelp dataset is about review rating prediction for restaurants where all methods achieve less than 37% accuracy: it is unclear how significant hypergraph modelling and the curated node features (e.g., latitude, longitude, city, state, etc.) are for rating prediction. The Walmart (and House) datasets do not come with node features, so it would be more compelling if hypergraph-only methods are used as baselines [e.g., Re-revisiting Learning on Hypergraphs: Confidence Interval and Subgradient Method, ICML'17].


3. **Positioning with Prior Work**

Experiments are restricted to transductive node classification in hypergraphs. The idea of $f_{\mathcal{V}\rightarrow\mathcal{E}}$ and $f_{\mathcal{E}\rightarrow\mathcal{V}}$ update rules have been successfully used in other tasks, e.g., (i) inductive text classification  [e.g., Be More with Less: Hypergraph Attention Networks for Inductive Text Classification, EMNLP'20], (ii) hyperlink prediction [e.g., Principled Hyperedge Prediction with Structural Spectral Features and Neural Networks], and (iii) recommendation [e.g., Session-based Recommendation with Hypergraph Attention Networks].
It would be interesting to see how AllSetTransformer and AllDeepSets would perform in these tasks.


### Update
After reading the responses to all the reviews, I have updated my recommendation from 5 -> 6

**Summary Of The Paper:**

Hypergraphs can capture group/set relationships in real-world data.
Several hypergraph neural networks have been proposed in the literature to exploit both group relationships among nodes and node features for learning with hypergraph data.
The contributions of the paper are
1) generalisation of most existing methods into a single framework (named AllSet),
2) exploration of AllSet based on Deep Sets [NeurIPS'17] and Set Transformer [ICML'19], and
3) empirical evaluation on existing benchmarks and three curated hypergraph datasets

**Summary Of The Review:**

Overall the paper is clear and of good quality in which claims are well supported by theoretical analysis. However, the novelty is incremental and experimental results are marginally significant. Positioning with missing prior work and evaluation on other hypergraph tasks would improve the paper.

---

> ### Author Response · Authors · 2021-11-14
> **Response to Reviewer LMVx (1/2)**
>
> We thank Reviewer LMVx for their positive comments regarding the quality and clarity of our paper. We address all questions raised below.
>
> ### Q1: Question on originality.
>
> Thank you for suggesting additional references, which we plan to include in our revision. Our equation (5) is similar to some results reported in very recent works. Indeed, we cited HNHN (Dong et al., 2020) and UniGNN (Huang & Yang 2021), and compared their findings to ours. However, we firmly believe that our work is the first to view $f_{\mathcal{V}\rightarrow \mathcal{E}}, f_{\mathcal{E}\rightarrow \mathcal{V}}$ as **multiset functions**. Our view of $f_{\mathcal{V}\rightarrow \mathcal{E}}, f_{\mathcal{E}\rightarrow \mathcal{V}}$ as multiset functions is crucial for the correct integration of Set Transformers and Deep Sets into our hypergraph learning framework. Hence, without the new interpretation of the propagation rules in terms of multiset functions, none of this work would have been possible and it confirms the importance of the main idea behind AllSet. Furthermore, within our AllSet framework, one can also use other more advanced multiset function learners such as Janossy pooling (Murphy et al. 2019), cited in our paper. Such a flexibility once again would not have been possible without the insight regarding multiset functions and their role in AllSet. We hope that the reviewer recognizes that this is a novel approach to hypergraph learning and that our work is not lacking in originality.
>
> ### Q2: Significance of empirical Evaluation.
>
> As mentioned in the original submission, many previous benchmarking datasets are too simple so that almost all methods offer comparable performance when applied to the same. This is one of the main reasons why we had to introduce new challenging datasets for hypergraph neural networks into our evaluation. The newly curated dataset Yelp is indeed very challenging as the best method (our AllSetTransformer) achieves less than 37% accuracy. Our performance gain over the best baseline method on Yelp (HGNN) is statistically significant. Note that there are 9 classes in Yelp dataset and the one with largest size has the portion of 26.28%. This means that the random guess will result in the accuracy of 26.28% which is apparently much lower than our AllSetTransformer performance. Including new challenging datasets should not be viewed as a drawback of our work but rather as a strong contribution to the field of hypergraph learning. For Walmart and House, we do agree that including a hypergraph-only method as a baseline can be more compelling. However, it is hard to compare the results of hypergraph neural networks with the hypergraph-only method, as our node features are ground truth labels corrupted with Gaussian noise.
>
> To address the concerns, we performed additional experiments using spectral clustering and label propagation with clique-expansion in place of hypergraph-only methods. For spectral clustering, we use the propagation matrix of HGNN described in equation (2), with weight $w_e=1$ as suggested in the HGNN paper. Since spectral clustering is an unsupervised method, we used a modification of Jonker-Volgenant algorithm with no initialization (implemented in scipy) to find the optimal assignment. For label propagation, we set the teleport probability $\alpha=0.5$, as suggested in the graph learning literature [5]. The results are as follows
>
> |          Accuracy (%)       | Walmart | House |
> |------------------------|---------|-------|
> | CE+Spectral clustering |     29.53    | 52.95  |
> | Label Propagation |     29.46    |  48.06 |
>
> Clearly, the additional simulation results strongly support our claim that the AllSet framework consistently outperforms almost all hypergraph learners on almost all datasets.
>
> Note that the reference provided by the reviewer does not include a public available code and we were not able to get the same from the authors. Furthermore, many existing hypergraph clustering methods are not able to deal with varying hyperedge sizes (and especially large hyperedges), such as Multilinear PageRank cited in our paper. This is the reason why we choose to use spectral clustering and label propagation with clique-expansion as the baseline of hypergraph-only methods.

---

> > ### Author Response · Authors · 2021-11-14
> > **Response to Reviewer LMVx (2/2)**
> >
> > ### Q3: Application to the other tasks.
> >
> > Thank you for suggesting other interesting applications, other than node classification. We will include relevant references suggested by the reviewer in our revision. This question is essentially the same as question Q3 of Reviewer Bdsv. For convenience, we restate our answer provided to Reviewer Bdsv once again.
> >
> > Although many other interesting tasks can be solved by using hypergraph neural networks, coming up with one architecture suitable for many different tasks is highly non-trivial and usually not done in the existing literature. For example, the state-of-the-art graph NNs for node classification, link prediction, and graph classification differ significantly. This can be easily observed from the OGB [1] leaderboard for each task. Furthermore, the corresponding tasks for hypergraphs are even more complicated to solve than those for graph. As a specific example, consider hyperlink prediction. Instead of computing similarities of two node representations, hyperlink prediction requires computing similarities of a larger number of node representations in hypergraphs, which is harder. The authors of Hyper-SAGNN (cited in our paper) proposed a special design to deal with hyperlink prediction tasks which is non-trivial to combine with our approach.
> >
> > Nevertheless, we do agree that it is interesting to apply the AllSet idea to different downstream tasks, and we plan to focus on the same in the future. At the same time, as already pointed out, our contributions to the node classification problem are significant standalone results.
> >
> > With regards to some of the references mentioned by Reviewer LMVx, we would like to point out that it is non-trivial to conduct experiments for these different tasks within the rebuttal period.
> >
> > Paper [2] is not based on PyTorch Geometric as our method is. Furthermore, each document in the proposed application requires one hypergraph to propagate on which is a very different setting compared to our node classification problem. Paper [3] basically tackles the hyperlink prediction problem. We have discussed how to address this question using Hyper-SAGNN as described above. Also, the implementation of [3] is not publicly available, nor is the one in [4].
> >
> >
> > ### Reference
> >
> > [1] Open Graph Benchmark: Datasets for Machine Learning on Graphs, Hu et al. NeurIPS 2020.
> >
> > [2] Be More with Less: Hypergraph Attention Networks for Inductive Text Classification, EMNLP'20
> >
> > [3] Principled Hyperedge Prediction with Structural Spectral Features and Neural Networks
> >
> > [4] Session-based Recommendation with Hypergraph Attention Networks
> >
> > [5] Combining Label Propagation and Simple Models Out-performs Graph Neural Networks, Huang et al. ICLR 2021.

---

> > > ### Comment · Reviewer_LMVx · 2021-11-15
> > > **Follow-up**
> > >
> > > Thanks for the response. There are still a couple of queries/comments.
> > >
> > > ---
> > >
> > > 1] **Positioning with Attention-based Models**
> > >
> > > The paper shows clearly that AllSet generalises HCHA  [1].
> > >
> > > From a theoretical perspective, because the multi-set functions are quite general, the paper would be stronger if AllSet was positioned with other attention-based neural networks on hypergraphs as well especially as some of them use rules of the form $f_{\mathcal{V}\rightarrow\mathcal{E}}$ and $f_{\mathcal{E}\rightarrow\mathcal{V}}$.
> > >
> > > In other words, can AllSet generalise published models such as HyperGAT [2], Hyper-SAGNN [2], and HGAT [4]?
> > >
> > > ---
> > >
> > > 2] **Evaluation on Datasets without Initial Node Features**
> > >
> > > I agree that it would be hard to compare the results with hypergraph-only methods if the node features are ground truth labels corrupted with Gaussian noise.
> > >
> > > Having said that, for (hyper)graphs without node features, the standard practice with (hyper)graph neural networks includes initialising nodes with (i) random multi-dimensional Gaussian features (number of dimensions can be tuned on the validation/development set), or (ii) one-dimensional unit features (i.e., all nodes are initialised to 1).
> > >
> > > Using one such (fair) feature initialisation for AllSet and comparing it with hypergraph-only methods (e.g., CE+Spectral clustering, Label Propagation as in the response) on Walmart and House would make the comparison more compelling.
> > >
> > > ---
> > >
> > > **References**
> > >
> > > [1] Hypergraph convolution and hypergraph attention. Pattern Recognition, 2021
> > >
> > > [2] Be More with Less: Hypergraph Attention Networks for Inductive Text Classification, EMNLP'20
> > >
> > > [3] Hyper-SAGNN: a self-attention based graph neural network for hypergraphs, ICLR'20
> > >
> > > [4] Session-based Recommendation with Hypergraph Attention Networks, SIAM International Conference on Data Mining 2021 (SDM'21).
> > >
> > > ---

---

> > > > ### Author Response · Authors · 2021-11-17
> > > > **Response to follow-up comments (1/2)**
> > > >
> > > > Thank you for your timely response. We addressed the follow-up comments below.
> > > >
> > > > ### Q1: Positioning with Attention-based Models
> > > >
> > > > Equation (3) in the HyperGAT [2] paper has a similar form as our AllSet Equation (5), except that the aggregator $AGGR_{node}$ does not make use of previously obtained representations (i.e. $f_j^{l-1}$ in Equation (3) of [2]). However, the authors of [2] define their aggregation function in a way that does not preserve the universal approximation properties of DeepSets and SetTransformers. For example, Equation (4) of [2] is just a weighted combination of linearly transformed features:
> > > >
> > > > $f_j^l = \sigma (\sum_{v_k \in e_j} \alpha_{jk} W_{1}h_k^{l-1} ).$
> > > >
> > > > Hence, the method cannot, in theory, approximate arbitrary multiset functions similar to HCHA [1]. The theory supporting Deep Sets clearly shows that we need the design of our Equation (7) in order to arrive at a universal approximator of multiset functions (i.e. the MLP-SUM-MLP structure). The Set Transformer architecture in our equation (8) can also model (7) but performs better in practice by leveraging attention mechanisms.
> > > >
> > > > In conclusion, our AllSet framework indeed generalizes the propagation rule of HyperGAT [2]. Of course, as mentioned in our previous response, there are more task-specific results presented in [2], as the authors focused on a problem different from ours.
> > > >
> > > > For Hyper-SAGNN [3], as already mentioned in our Related Work section, it is not easy to determine if it can be modeled by our AllSet framework (we plan to investigate this interesting question as part of our future work). The propagation rule in equation (5) of [3] can be modeled by our AllSet rules. However, there are more sophisticated processing units in Hyper-SAGNN, such as the Hadamard power of the resulting node representations. This is to be expected as Hyper-SAGNN focuses on hyperlink prediction and not on node classification like we do.
> > > >
> > > > For HGAT [4], the propagation rules defined in Equations (3.1) and (3.3) can be modeled by our AllSet framework. In fact, HGAT is very similar to HyperGAT [2] and the two papers share a number of contributing authors. We again emphasize that HGAT [4] has its own task-specific designs which make the model hard to compare to our AllDeepSets or AllSetTransformer. We believe that these task-specific designs represent interesting contribution in their own right and we do not want to simply claim that all results in [2],[3], and [4] are special instances of our findings.

---

> > > > > ### Author Response · Authors · 2021-11-17
> > > > > **Response to follow-up comments (2/2)**
> > > > >
> > > > > ### Q2: Evaluation on Datasets without Initial Node Features
> > > > >
> > > > > We want to emphasize again that one key benefit of GNNs for node classification tasks over graph-only methods is that they can jointly leverage the information from node features and the graph topology. This is also the main reason why we introduced synthetic node features, as we wanted to test the ability of different hypergraph neural networks to combine both types of information. We believe that using constant or completely random node features is common for link prediction and graph classification tasks, but not node classification. We do not see any problems related to using synthetic node features to test hypergraph neural networks, as every model has the same input.
> > > > >
> > > > > Nevertheless, to address the concern of the reviewer, we also conducted a new set of  experiments to compare AllSet with hypergraph-only methods. Instead of using constant or completely random node features, we propose to use our synthetic node features for nodes within the **training set only**. Node features in the validation and test set are replace by all-zero vectors. Note that we also find that our previous results (provided in the response above) did not use multiple random splits within our experimental setting for hypergraph neural networks. So, we modified our code and repeated the experiments for 10 runs. The new result is as follows:
> > > > >
> > > > > |  | Walmart (0.6) | House (0.6) |
> > > > > |:---:|:---:|:---:|
> > > > > | CE+Spectral clustering | 29.54±0.23 | 51.27±1.57 |
> > > > > | CE+Label Propagation | 34.49±0.34 | 48.82±2.27 |
> > > > > | AllSetTransformer (using only node features in training set) | 36.01±1.96 | 51.76±2.33 |
> > > > >
> > > > > We can see that for both datasets, our AllSetTransformer offers comparable or better performance than hypergraph-only approaches, even in a setting that is not completely suitably for analysis with hypergraph neural networks. Note that having a similar performance as that of hypergraph-only methods in this case does not mean that hypergraph-only methods are better than hypergraph neural networks, as the latter has the flexibility to leverage potentially useful node feature information. This is also the standard setting for node classification, which is the main focus of this paper.
> > > > >
> > > > > ### References
> > > > >
> > > > > [1] Hypergraph convolution and hypergraph attention. Pattern Recognition, 2021
> > > > >
> > > > > [2] Be More with Less: Hypergraph Attention Networks for Inductive Text Classification, EMNLP'20
> > > > >
> > > > > [3] Hyper-SAGNN: a self-attention based graph neural network for hypergraphs, ICLR'20
> > > > >
> > > > > [4] Session-based Recommendation with Hypergraph Attention Networks, SIAM International Conference on Data Mining 2021 (SDM'21).

---

> > > > > > ### Comment · Reviewer_LMVx · 2021-11-24
> > > > > > **Thanks for the response**
> > > > > >
> > > > > > Thanks to the authors again for the response. I have decided to increase my rating by a point after reading the author responses to all reviews.

---

### Official Review · Reviewer_unoj · 2021-11-03

**Correctness:** 4
**Technical Novelty And Significance:** 3
**Empirical Novelty And Significance:** 3
**Recommendation:** 8
**Confidence:** 3

**Main Review:**

Strengths:

- Overall, I think this is a well-organized paper and I enjoyed reading it. The motivations for the model design are well discussed as is the related work.
- The proposed idea is simple and neat. Many hypergraph Laplacian-like mappings have been considered and this gives a possible data-centric answer to the question "which one should I use?"
- The experiments consider a wide range of hypergraph datasets, which is somewhat rare in hypergraph classification literature. Moreover, the authors have collected several new hypergraph datasets, which I believe will be very useful for the community

Weaknesses:
- The experiments are not completely clear to me. In particular: (1) I think you should specify more clearly what is the depth of MLP and Set Transformer layers in AllDeepSets and AllSetTransformer and (2) you should more clearly specify what is the overall amount of known labels you used in the experiments and what is the amount used for training the AllSets architectures
- The notation in equation (8) should be better specified (e.g. what is LN or || )
- Other propagations which are "in-between" CE and tensors have been considered recently, e.g. [1-3]. Would these be particular cases of your proposed framework too?
[1] Tudisco et al - Node and edge nonlinear eigenvector centrality for hypergraphs
[2] Arya et al - Adaptive Neural Message Passing for Inductive Learning on Hypergraphs
[3] Prokopchik - A nonlinear diffusion method for semi-supervised learning on hypergraphs


**Summary Of The Paper:**

Transferring standard graph operators to the hypergraph setting is non-trivial and several message passing operators have been considered in the setting of hypergraphs, including clique-expansion and tensor based. In this paper the authors propose a general framework where learnable multiset functions are used in order to learn the hypergraph propagation map from the data. In particular, this framework contains  many previously considered propagation maps as particular cases

**Summary Of The Review:**

Overall, this is a solid paper which proposes one simple, yet novel and efficient idea to generalize propagation maps on hypergraphs. Solid experimental results are provided in the paper.

I like this paper and I would like to see it accepted. I will raise my score if the weaknesses are properly addressed

-----

I am happy with the feedback provided to my concerns and those of other reviewers so I will raise my score

---

> ### Author Response · Authors · 2021-11-14
> **Response to Reviewer unoj**
>
> We thank Reviewer Bdsv for their valuable time and helpful comments. We address all the question raised below.
>
> ### Q1: Experiment details.
>
> Due to the space limitations, we had to relegate most of the details pertaining to our experiments to Appendix I. Here, we repeat some of them and add additional explanations, both of which will be included in the revision. Specifically, the depth of MLP in AllSetTransformer and AllDeepSets is 2. We use one layer (a full $\mathcal{V}\rightarrow\mathcal{E}\rightarrow \mathcal{V}$ propagation rule layer) for AllSetTransformer and AllDeepSets. This means we use 2 Set Transformer layers for AllSetTransformer (one for $\mathcal{V}\rightarrow\mathcal{E}$ and one for $\mathcal{E}\rightarrow \mathcal{V}$). The split ratio of labels for train/val/test sets is (50 %/ 25 %/ 25 %), applied on all datasets (please see the first paragraph of Section 6).
>
> ### Q2: Notations in equation (8).
>
> LN stands for Layer Normalization which is specified right after equation (8). || stands for concatenation, as defined in Section 2 where we introduced the relevant notation. We will make these explanations more transparent in our revision.
>
> ### Q3: Discussion on propagations “in-between” CE and tensor-based.
>
> Great question! We thank Reviewer unoj provide these very recent references. We will cite them in our revision, and explain their relationship to our AllSet method as detailed below.
>
> In [1], the authors studied node and edge nonlinear eigenvector centrality. Our AllSet framework formulation can be made to include propagation rules based on eigenvector centrality (equation (2) in [1]). The differences are that the Tudisco et al. have to constrain the vectors (x,y in [1]) to be entry-wise positive due to the nature of “centrality”. In fact, if we consider the 4 mappings in [1] ($f,g,\phi,\psi$) to be represented by MLPs and drop the positivity constraint, then we arrive at a AllDeepSet formulation with weighted summations involving ($w(e),v(i)$). These weights can be modeled with our AllSetTransformer. Instead of studying a data-centric method to learn these mapping, Tudisco et al. focus on theoretical properties of a specific family of mappings and how to compute the centrality vector based on these mappings. In fact, we did cite an earlier paper by the same author (Tudisco et al 2020) as it motivated our work.
>
> Regarding [2], we actually did cite and discuss the results of an earlier version of the paper (HyperSAGE, Arya et al. 2020). The generalized mean aggregation approach (equation (4) and (5)) can also be included within our AllSet framework, as proved in our Theorem 3.4.
>
> In [3], the authors also discuss nonlinear diffusion, similar to [1], but in the context of a semi-supervised node classification problem. Their diffusion map $\Phi$ (equation (5)) is also covered by AllSet. If one chooses the mixing mappings $\sigma,\rho$ to be MLPs, then equation (5) in [3] is again AllDeepSets with a weighted summation. It can also be modeled by our AllSetTransformers. Note that the authors of [3] also focus on the theoretical analysis of a specific family of mixing mappings, while we focus on data-centric methods for learning general propagation rules. We think that one can also view the work in [3] as a hypergraph version of SGC [4]. Since the propagation part in [3] is fixed, the underlying computational complexity is better than that of our AllSetTransformer and AllDeepSets. But then, our AllSet methods can learn more complex propagation rules in an adaptive way. Also, we believe the theoretical analysis in [1,3] can be helpful with regards to the interpretability of AllSetTransformer. These discussions also point out an interesting future direction brought up in comment Q1 of Reviewer Bdsb. We thank Reviewer unoj once again for suggesting this discussion, which will improve the content of our manuscript.
>
> ### References
> [1] Tudisco et al - Node and edge nonlinear eigenvector centrality for hypergraphs
>
> [2] Arya et al - Adaptive Neural Message Passing for Inductive Learning on Hypergraphs
>
> [3] Prokopchik - A nonlinear diffusion method for semi-supervised learning on hypergraphs
>
> [4] Simplifying graph convolutional networks, Wu et al. ICML 2019.

---

> ### Author Response · Authors · 2021-11-25
> **Reply to Reviewer unoj**
>
> Dear Reviewer unoj,
>
> Thank you again for your valuable comments. We are wondering if your concerns have been addressed properly. Please let us know if you have any further questions after reviewing our answers.
>
> Best regards,
>
> The authors

---

### Official Review · Reviewer_Bdsv · 2021-11-04

**Correctness:** 3
**Technical Novelty And Significance:** 3
**Empirical Novelty And Significance:** 2
**Recommendation:** 6
**Confidence:** 3

**Main Review:**

Strengths of the paper:

S1. The problem is well-motivated, with significant coverage of existing work based on clique expansion and tensor-based propagation.

S2. Experiments include a significant number of baselines (ten models), as well as on ten different datasets.

S3. There are theoretical analyses showing how the proposed framework is a generalisation of some existing works.


Weaknesses of the paper:

W1. As the proposed framework is in some sense a generalisation or a unification of existing propagation mechanisms, such as clique expansion and tensor-based propagation, there should be an ablation analysis to understand the contribution of each component.

W2. The experimental discussion are focused on just showing numerical comparisons.  A more discursive analysis that explains why the method works better than the existing methods would help to reveal where the sources of improvements are.

W3. Since the core is to derive hyper graph representations, experimenting with a single task is rather limited.  It would be good to have additional evaluation tasks beyond semi-supervised classification.

**Summary Of The Paper:**

The paper proposes a framework for hyper graph neural networks.  It argues that existing work propagates a hyper graph by first transforming it into a regular graph trough clique expansion, which may lose information.  Alternatively, the propagation is based on tensors.  The proposed framework seeks to combine these different ways of propagations into one unified framework.  The proposed framework is called AllSet, with two instantiations AllDeepSets and AllSetTransformer.  Experiments are conducted on several graph datasets.

**Summary Of The Review:**

The paper proposes a framework for hyper graph neural networks, which is an interesting problem.  Though rigorous in its treatment and experiments, further analysis on what makes the work better then the baselines would help in better appreciating the significance of the contributions.

---

> ### Author Response · Authors · 2021-11-14
> **Response to Reviewer Bdsv**
>
> We thank Reviewer Bdsv for their valuable time and comments. We address all the question raised below.
>
> ### Q1: “As the proposed framework is in some sense a generalization or a unification of existing propagation mechanisms, such as clique expansion and tensor-based propagation, there should be an ablation analysis to understand the contribution of each component.”
>
> Great question! As Reviewer Bdsv pointed out, our AllSet framework contains clique-expansion-based and tensor-based propagation as special cases. However, the propagation rule learned by our AllSet framework is not necessarily a “weighted combination” of these two. Instead, it can produce results that are very different from those generated by either method. Hence, an ablation study of the type usually performed in the ML literature may not be possible and also hard to perform in our case. Nevertheless, we did indeed compare AllSet with clique-expansion-based methods (such as CEGCN, CEGAT) in Table 2 of the original submission. We also tried to implement a hypergraph neural network based on Zprop. Unfortunately, Zprop is known to have the “unit problem” and its product formalism causes numerical instabilities when large hyperedges are present. This is discussed in our paper right below equation (4).
>
> Reviewer unoj had a similar question (Q3), pertaining to clique-expansion and tensor-based method. Please refer to our response to that question as well.
>
>
> ### Q2: “The experimental discussion is focused on just showing numerical comparisons. A more discursive analysis that explains why the method works better than the existing methods would help to reveal where the sources of improvements are.”
>
> Thanks for the suggestion. Indeed, we have also considered doing some discursive analysis. We originally planned to examine the weights learned by our AllSetTransformer to determine what kind of propagation rules are learned for each dataset. However, we ran into interpretability issues that are highly non-trivial as there are many weights to analyze and “obvious” patterns are discernable. We still plan to investigate this direction in greater depth in our future work, by including new datasets into the analysis, and developing specialized learning methods for identifying propagation rules based on the weights.
>
> This being said, we did perform a comparative analysis of DeepSets and Set Transformers, both of which are universal approximators of the general multiset functions. Our results indicate that AllSetTransformer can learn the relevant multiset functions better than AllDeepSets. This confirms the findings of Lee et al. 2019 that attention mechanisms are crucial for learning multiset functions in practice. We referred to this explanation in the last paragraph of our paper. The same paragraph also includes a discussion of the particular problems associated CE-based methods when there are large hyperedges and the instability of the mediator approach in HyperGCN. All these were included in addition to the numerical results, but were not elaborated on to a greater extent due to space limitations and the fact that some explanations were already provided by the original paper proposing the propagation methods.
>
> ### Q3: “Since the core is to derive hypergraph representations, experimenting with a single task is rather limited. It would be good to have additional evaluation tasks beyond semi-supervised classification.”
>
> Although many other interesting tasks can be solved by using hypergraph neural networks, coming up with one architecture suitable for many different tasks is highly non-trivial and usually not done in the existing literature. For example, the state-of-the-art graph NNs for node classification, link prediction, and graph classification differ significantly. This can be easily observed from the OGB [1] leaderboard for each task. Furthermore, the corresponding tasks for hypergraphs are even more complicated to solve than those for graph. As a specific example, consider hyperlink prediction. Instead of computing similarities of two node representations, hyperlink prediction requires computing similarities of a larger number of node representations in hypergraphs, which is harder. The authors of Hyper-SAGNN (cited in our paper) proposed a special design to deal with hyperlink prediction tasks which is non-trivial to combine with our approach.
>
> Nevertheless, we do agree that it is interesting to apply the AllSet idea to different downstream tasks, and we plan to focus on the same in the future. At the same time, as already pointed out, our contributions to the node classification problem are significant standalone results.
>
>
> ### Reference
> [1] Open Graph Benchmark: Datasets for Machine Learning on Graphs, Hu et al. NeurIPS 2020.

---

> ### Author Response · Authors · 2021-11-25
> **Reply to Reviewer Bdsv**
>
> Dear Reviewer Bdsv,
>
> Thank you again for your great comments. We are wondering if your concerns have been addressed properly. Please let us know if you have any further questions after reviewing our answers.
>
> Best regards,
>
> The authors

---

### Official Review · Reviewer_6wpr · 2021-11-08

**Correctness:** 4
**Technical Novelty And Significance:** 3
**Empirical Novelty And Significance:** 3
**Recommendation:** 8
**Confidence:** 4

**Main Review:**

This paper proposes a generalization of the hypergraph neural networks using multiset functions. Instead of doing clique expansion as in many existing methods, the proposed method learns explicit hypergraph representations, and uses two multiset functions to perform message propagations, one from nodes to hyperedge and another from hyperedge to nodes. The method is shown to be a general framework that encompasses existing GNN methods. Experimental results show that the AllSetTransformer achieves competitive results or outperforms existing methods on a large set of benchmark datasets.

The proposed method unifies various methods for hypergraph neural networks, and gives a competitive formulation across a wide spectrum of problems. This is a nice contribution to the community.


**Summary Of The Paper:**

This paper proposes a generalization of the hypergraph neural networks using multiset functions.

**Summary Of The Review:**

The paper makes a nice contribution to the community.

---

> ### Author Response · Authors · 2021-11-14
> **Response to Reviewer 6wpr**
>
> We thank Reviewer 6wpr for their positive comments. Please let us know if you have any further questions. We will address them as soon as possible.

---

### Decision · Program_Chairs · 2022-01-20

**Decision:**

Accept (Poster)

**Comment:**

This paper proposes a hypergraph representation learning based on multiset encoding, which  covers most existing propagation methods for hypergraph neural networks. The authors provide theoretical proofs that both CE-based and tensor-based propagation rules can be represented as a composition of two multiset functions, and propose two different multiset encoding functions, based on DeepSets and SetTransformer. The authors validate their method for its semi-supervised node classification performance on multiple benchmark datasets, showing that it is superior or comparable to a large number of existing works on hypergraph representation learning.

The following is the summary of the pros and cons of the paper mentioned by the reviewers:

Pros
- The proposed framework generalizes existing message passing methods for hypergraphs, and authors provide theoretical proofs on how it can generalize to two different types of propagation rules for hypergraph representation learning.
- The paper is well-organized and is clearly written, and the code is provided for reproduction.
- The experiments consider a wide range of hypergraph datasets and baselines, and the proposed method either outperforms them or at least achieves comparable performance.

Cons
- It is still unclear where the benefits come from, due to lack of ablation studies and deeper analysis.
- Experiments are only restricted to the semi-supervised node classification task.

While the initial reviews were split due to these negative points, all reviewers unanimously recommended for acceptance after the discussion period, as they found the responses from the authors satisfactory.

In summary, this is a well-written paper that proposes a neat framework for hypergraph representation learning that generalizes to most existing methods, backed up by compelling performance on benchmark datasets, which will make it a strong addition to ICLR. However, as mentioned by the reviewers there should be more ablation studies and in–depth analysis of what makes the proposed multiset function more effective, as this is lacking in the current version of the paper. It would be worthwhile to also validate the proposed framework on other tasks (e.g. graph classification tasks).

One minor thing that I want to point out is regarding the claim that this is the first attempt to connect the problem of learning multiset functions with hypergraph neural networks.  [Jo et al. 21], which is a hypergraph-based framework for edge representation learning, utilized GMT [Baek et al. 21], which performs multiset encoding using SetTransformer for hypergraph representation learning, and thus I suggest the authors to tone down on the claim that this is the first work that connects multiset encoding with hypergraph neural networks, and properly acknowledge this.

[Jo et al. 21] Edge Representation Learning with Hypergraphs, NeurIPS 2021